# Uncertainties in Physics-informed Inverse Problems:
# The Hidden Risk in Scientific AI

## Abstract

Physics-informed machine learning (PIML) integrates partial differential equations (PDEs) into machine learning models to solve inverse problems, such as estimating coefficient functions (e.g., Hamiltonians) that characterize physical systems. While such functions are often learned by optimizing predictive performance, physical modeling requires criteria beyond prediction accuracy to identify physically meaningful solutions. In this work, we propose a framework to quantify and analyze structural uncertainty in the estimation of PDE coefficient functions within PIML. We demonstrate the framework on a reduced magnetohydrodynamics model and show that coefficient functions can be non-unique under purely predictive learning, whereas imposing appropriate geometric constraints enables unique and physically consistent identification.

## 1. Introduction

There is active research on elucidating physical laws in a data-driven manner using machine learning (Karniadakis et al., 2021; Hao et al., 2022). When the basis functions of physical laws are partially known, governing equations can be extracted from time-series data via linear regression (Brunton et al., 2016) or symbolic regression (Udrescu & Tegmark, 2020). Even with limited prior information, expressive models such as deep neural networks (DNNs) have been combined with explainable AI (XAI) techniques (Barredo Arrieta et al., 2020; Love et al., 2023) to obtain interpretable physical information, including interaction laws (Cranmer et al., 2020) and conservation laws (Kaiser et al., 2018; Wetzel et al., 2020; Liu & Tegmark, 2021; Ha & Jeong, 2021; Liu & Tegmark, 2022; Mototake, 2021; Zhang et al., 2021; Lu et al., 2023).

[1]Anonymous Institution, Anonymous City, Anonymous Region, Anonymous Country. Correspondence to: Anonymous Author <anon.email@domain.com>.

Preliminary work. Under review by the International Conference on Machine Learning (ICML). Do not distribute.

Whereas machine learning models are typically evaluated based on predictive performance, physics does not rely solely on prediction accuracy to assess model validity. A canonical example is the historical competition between the geocentric and heliocentric models: despite comparable predictive accuracy given the observational data available before Kepler, the heliocentric model was ultimately favored because it provided a more physically coherent interpretation of planetary motion (Kuhn, 1957; Principe, 2011; Weinberg, 2015). From the perspective of data-driven modeling, this episode illustrates that model selection based purely on predictive performance can lead to non-unique and potentially misleading physical interpretations. If modern machine learning methods, such as physics-informed machine learning (PIML) methods (Karniadakis et al., 2021; Hao et al., 2022), were applied to such historical data, they could plausibly favor one of multiple competing models, creating a risk of drawing physically incorrect conclusions (Fig. 1). This study focuses not on XAI techniques themselves, but on the inherent uncertainty residing in the physical model.

Uncertainties in data-driven modeling can be classified into three types (Pelz et al., 2021). *Structural uncertainty* persists even with infinite, noiseless data when multiple model structures are consistent with the same observations. *Model-form uncertainty* arises when the true physical law lies outside the assumed model class. *Data uncertainty* is caused by finite or noisy data, even for identifiable models. The appropriate strategy for addressing uncertainty depends on its type: structural uncertainty requires additional physical constraints, model-form uncertainty calls for improved modeling, and data uncertainty can be addressed through statistical approaches such as Bayesian inference (Pelz et al., 2021; René & Longtin, 2025). Among these, structural uncertainty is the most fundamental, as it is an inherent property of the physical model itself and should be evaluated prior to other uncertainty analyses.

Partial differential equations (PDEs) are among the most common representations of physical models. In PDE-based systems, coefficient functions play a central role in governing physical behavior; for example, in Hamiltonian systems, the Hamiltonian function acts as a coefficient function that determines the dynamics of observable variables. In PIML, approaches such as physics-informed neural net-

works (PINNs) (Raissi et al., 2019; Adams-Tew et al., 2024; Depina et al., 2022; Sahin et al., 2024; Yang et al., 2021) and Hamiltonian neural networks (HNNs) (Schmidt & Lipson, 2009; Greydanus et al., 2019a; Toth et al., 2019; Bondesan & Lamacraft, 2019) estimate such coefficient functions by enforcing PDE constraints during training. Although this framework enables data-driven modeling of complex physical systems, coefficient functions are typically inferred by optimizing predictive performance, which can obscure structural non-uniqueness. Since structural uncertainty arises from the PDE itself rather than from the learning algorithm, it is both possible and necessary to assess it independently of machine learning.

The ill-posedness of inverse problems, particularly non-uniqueness, has been extensively studied in mathematical inverse theory, including Calderón-type problems (Calderón, 1980; Sylvester & Uhlmann, 1987) and hyperbolic inverse problems analyzed via Carleman estimates (Yamamoto, 2009; Bellassoued & Yamamoto, 2017). In the machine learning context, it has been shown that PINNs may converge to physically incorrect solutions despite low training loss (Krishnapriyan et al., 2021), and related uncertainty analyses have highlighted epistemic structural uncertainty in inverse modeling without adequate constraints (Mishra et al., 2022). These studies, based on analyses of specific classes of PDEs, suggest both the existence of structural uncertainty and the importance of analyzing such uncertainty prior to applying machine learning methods for understanding physical phenomena. Therefore, developing a general framework applicable to a wide range of physical models (PDEs) would strongly support data-driven research in physics.

The purpose of this study is to develop a framework to quantitatively evaluate structural uncertainty in data-driven estimation of PDE coefficient functions.

## 2. Related Works

### 2.1. PIML and Inverse Problem

PIML is an emerging framework that integrates physical laws, such as PDEs, into the training process of machine learning models such as DNNs. Prominent instantiations of this idea include PINNs (Raissi et al., 2019) and HNNs (Greydanus et al., 2019b), where a neural network is trained not only to fit observational data but also to satisfy a given PDE.

In the PINN framework, the governing PDE is typically of the form $\mathcal{N}[u, a](x) = 0$, $x \in \mathbb{R}^d \times [0, T]$, where $\mathcal{N}$ is a nonlinear differential operator derived from physical laws, $u : \mathbb{R}^d \to \mathbb{R}$ is a sufficiently smooth function of observation values, $a : \mathbb{R}^d \to \mathbb{R}$ is a sufficiently smooth coefficient function of PDE, and $x := (x, t) \in \mathbb{R}^d \times [0, T]$. For ex-

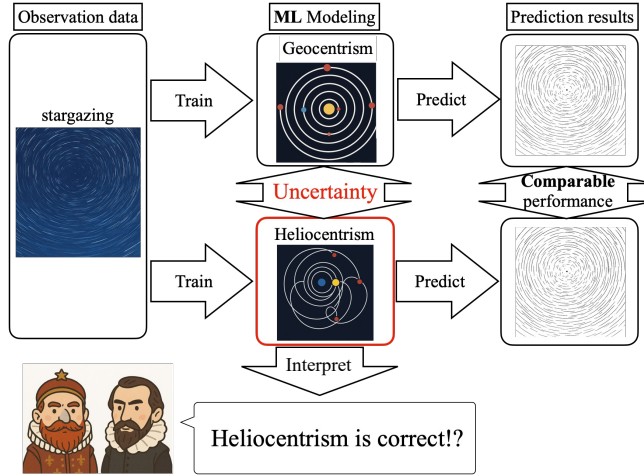

*Figure 1.* Risks of scientific research using machine learning. In the presence of an uncertainty, a machine learning model may sometimes provide an interpretation that is physically unfavorable.

ample, the following equations are included: $\mathcal{N}[u, a] := u(x) - \nabla \cdot (a(x) \nabla u(x)) = 0$. To enforce this PDE constraint in machine learning, the loss function used in training is augmented by a physics-informed term:

$$\mathcal{L}(\theta_u, \theta_a) = \mathcal{L}_{\text{data}}(\theta_u) + \lambda_{\text{PDE}} \mathcal{L}_{\text{PDE}}(\theta_u, \theta_a), \quad (1)$$

where $\mathcal{L}_{\text{data}}(\theta_u) = \frac{1}{N} \sum_{i=1}^{N} \|u_\theta(x_i) - u_i\|^2$, $\mathcal{L}_{\text{PDE}}(\theta_u, \theta_a) = \frac{1}{N_r} \sum_{r=1}^{N_r} \|\mathcal{N}[u_{\theta_u}, a_{\theta_a}](x_r)\|^2$, where $u_{\theta_u}(x)$ and $a_{\theta_a}(x)$ are neural network models parametrized by $\theta$. Here, $\{(x_i, u_i)\}_{i=1}^{N}$ are supervised data points, whereas $\{(x_r)\}_{r=1}^{N_r}$ are residual points where the PDE is enforced. The parameter $\lambda_{\text{PDE}}$ balances the relative importance of data fidelity and physics conformity. In this framework, the partial differential coefficient function $a(x, t)$ can also be estimated by minimizing $\mathcal{L}(\theta_u, \theta_a)$. In the HNN-type framework, the objective is not to estimate the observation function $u(x)$, but to estimate the partial differential coefficient function $a(x)$. Thus, the loss function is given as $\mathcal{L}_{\text{HNN}}(\theta) = \frac{1}{N} \sum_{i=1}^{N} \|\mathcal{N}[u, a_\theta](x_i)\|^2$, where $\{(x_i, u_i)\}_{i=1}^{N}$ are given as supervised data. From there, the partial derivatives of $u(x)$ in PDE, $\mathcal{N}[u, a_\theta](x_i)$, are assumed to be given numerically. For example, if the PDE is a canonical equation of motion, the loss function is given by

$$\mathcal{L}_{\text{HNN}}(\theta) = \frac{1}{N} \sum_{i=1}^{N} \left\| \begin{bmatrix} \frac{\partial H_\theta}{\partial p}(q_i, p_i) \\ -\frac{\partial H_\theta}{\partial q}(q_i, p_i) \end{bmatrix} - \begin{bmatrix} \dot{q}_i^{\text{obs}} \\ \dot{p}_i^{\text{obs}} \end{bmatrix} \right\|^2, \quad (2)$$

where the observation function is $u(t, q, p) = (t, q, p)$ and the coefficient function is $a(t, q, p) = H(q, p)$. In HNNs, the coefficient function $a(x)$ is estimated by minimizing the loss function $\mathcal{L}_{\text{HNN}}(\theta)$ similar to that of PINNs.

PIML has been successfully demonstrated in various tasks, including forward simulation, spatiotemporal forecasting (Karniadakis et al., 2021), and inverse problems such as parameter estimation (Raissi et al., 2018). For inverse problems, the physical constraint often compensates for limited data, enabling the estimation of unknown coefficient functions or source terms. However, the learned solution may not be unique: the PDE residual can be small even when multiple, distinct functions explain the data equally well using the same physical model.

Recent studies have highlighted the lack of identifiability guarantees in PINNs (Yang et al., 2021). In particular, the minimization of $\mathcal{L}_{\mathrm{PDE}}$ does not necessarily imply that the estimated parameters or functions are physically meaningful or unique. Furthermore, the structure of the differential operator $\mathcal{N}$, the available observation data, and the expressivity of the neural network all affect the identifiability and uncertainty of the learned solution. These findings emphasize the need for a rigorous theoretical framework for understanding the ill-posedness and uncertainty inherent in physics-informed inverse problems.

### 2.2. Types of Uncertainty and Our Focus

Following Pelz et al. (Pelz et al., 2021), we distinguish three main types of uncertainty that arise in physics-informed modeling. The first is *structural uncertainty*, which refers to situations in which multiple models or coefficient functions are exactly consistent with the same input–output relation induced by the true governing equations. It persists even in the idealized limit of infinite, noise-free data and reflects non-uniqueness inherent to the inverse problem itself. The second is *model-form uncertainty*, which arises when the true physical law lies outside the assumed model class or PDE family, so that no choice of parameters within the model can fully explain the observations. The third is *data uncertainty*, which encompasses finite-sample effects and measurement noise, and affects parameter estimation even when the model class is correct.

Importantly, these three types of uncertainty call for fundamentally different strategies for mitigation. Structural uncertainty must be addressed by introducing additional physical constraints or complementary information to enforce uniqueness, whereas model-form uncertainty requires refinement or extension of the underlying model class. In contrast, data uncertainty is typically handled through statistical or Bayesian approaches that explicitly account for finite data and noise. Table 1 summarizes these types of uncertainty, their typical sources, and representative mitigation strategies.

In this work, we focus on *structural uncertainty*. A canonical example is gauge invariance in Maxwell's equations. Such uncertainty reflects an intrinsic property of the physical system and persists independently of data quantity or quality. Because structural uncertainty originates from the governing physical laws, it should be identified and evaluated prior to other types of uncertainty.

### 2.3. Structural Uncertainty in PIML

In the inductive estimation of PDE coefficient functions addressed by PIML, there have been several studies that analyze the degree and structure of structural uncertainty. However, most of these studies have been conducted under specific assumptions or for particular classes of PDEs. Below, we briefly review related prior work. Classical studies, such as Calderón's problem (Calderón, 1980) and its resolution for elliptic equations (Sylvester & Uhlmann, 1987), established theoretical foundations for evaluating uncertainty in coefficient functions under full-boundary measurement assumptions. For hyperbolic equations, Carleman estimates have played a central role in deriving conditional uniqueness and stability results under geometric constraints (Yamamoto, 2009; Bellassoued & Yamamoto, 2017). These theoretical works have substantially advanced the understanding of uncertainty for specific classes of PDEs, while their scope is inherently limited to particular equation types.

From the perspective of machine-learning-based approaches, there are also studies that focus on uncertainty and non-uniqueness in coefficient-function estimation. Krishnapriyan et al. (Krishnapriyan et al., 2021) demonstrated that, in PINNs, flat or multimodal optimization landscapes can lead to physically inappropriate solutions even when the loss value is small. Mishra et al. (Mishra et al., 2022) further analyzed such inverse problems from the viewpoint of uncertainty quantification, highlighting the intrinsic uncertainty that arises when physical constraints are insufficient. These studies collectively underscore the difficulty of inverse problems in PIML and the importance of uncertainty evaluation.

Bayesian extensions, such as B-PINNs (Yang et al., 2021; Mishra et al., 2022), provide a probabilistic framework by introducing distributions over unknown quantities and inferring posterior distributions. While these approaches enable qualitative assessments of uncertainty, they are not designed to fundamentally distinguish among the three types of uncertainty discussed above.

Taken together, prior studies have highlighted the importance of evaluating uncertainty in coefficient-function estimation from different perspectives. As discussed earlier, the three types of uncertainty require fundamentally different strategies for mitigation, making it essential to assess them in a disentangled manner. Therefore, methods that isolate structural uncertainty and quantitatively characterize its degree and structure remain an important and open problem in the study of uncertainty evaluation for coefficient-function

*Table 1.* Types of uncertainty in inverse estimation of physics model.

| Type | Source | Typical remedy |
|------|--------|----------------|
| Structural | Multiple models fit the same data and PDE | Add physical constraints or new data types |
| Model-form | True law outside the assumed model/PDE class | Enrich the model class, model selection |
| Data | Finite/noisy observations | Bayesian/statistical modeling, regularization |

estimation.

## 3. Theoretical Preliminaries for Uncertainty Evaluation

**Definition 1** (Structural Uncertainty under a PDE Constraint). Let $a : \Omega \to \mathbb{R}$ denote the true underlying function, and let $u^\star$ be the corresponding state satisfying $\mathcal{F}(u^\star, a) = 0$ on $\Omega$. We say that *structural uncertainty* with respect to $a(x)$ arises if there exists another function $\hat{a}(x) \not\equiv a(x)$ such that $\mathcal{F}(u^\star, \hat{a}) = 0$ on $\Omega$.

**Definition 2** ($k$-Leaf Set of Partial Derivatives). Let $a : \Omega \to \mathbb{R}$ be a sufficiently smooth function defined on an open domain $\Omega \subset \mathbb{R}^d$. Using the multi-index notation $\alpha = (\alpha_1, \ldots, \alpha_d) \in \mathbb{N}^d$, we obtain the arbitrary partial differential coefficient of $a$ as $|\alpha| := \sum_{i=1}^{d} \alpha_i, \quad \partial^\alpha a(x) := \frac{\partial^{|\alpha|} a}{\partial x_1^{\alpha_1} \cdots \partial x_d^{\alpha_d}}$. Let $A_k := \{\alpha \in \mathbb{N}^d \mid |\alpha| = k\}$ be the set of all multi-indices of the total order $k$. Then, the $k$-*leaf set of partial derivatives* $S_k^{\text{leaf}}$ is defined as $S_k^{\text{leaf}} := \{\partial^\alpha a(x) \partial^\beta a(x) \mid \alpha \in A_k, |\beta| = 1\}$.

Example: If $\{\partial^\alpha a(x) \mid \alpha \in A_1\}$ is given by $\{\partial_x a(x), \partial_y a(x)\}$, then the 2-leaf set of partial derivatives is $\{\partial_{xx} a(x), \partial_{xy} a(x), \partial_{yx} a(x), \partial_{yy} a(x)\}$.

**Theorem 1** (Uniqueness of Coefficient Function up to Polynomial under Root Derivative Information). Let $u : \mathbb{R}^d \to \mathbb{R}$ and $a : \mathbb{R}^d \to \mathbb{R}$ be sufficiently smooth functions, and consider $m$-th PDEs of the form

$$\sum_{\alpha \in A_{\geq m}} \varphi_\alpha^{(\ell)}(x, u(x), \partial u(x), \partial^2 u(x), \ldots) \cdot \partial^\alpha a(x) = C^{(\ell)},$$

where $C^{(\ell)}$ is constant, $\ell = 1, \ldots, L$, and $\varphi_\alpha^{(\ell)}$ is an arbitrary scalar function, $A_{\geq m} := \{\alpha \mid m \leq |\alpha|\}$, $m \geq k$, and $\partial^k u(x)$ represents the arbitrary set of $k$-th-order partial differential coefficients. That is, PDEs are linear in $\partial^\alpha a(x)$. Assume that PDEs have a $k$-leaf set $S_k^{\text{leaf}}$ in their equations. Discretize $\Omega$ on an infinitesimal grid with spacing $\varepsilon > 0$, and denote the grid points as $x^{(i)} \in \mathbb{R}^d$ for $i = 1, \ldots, N$. For each grid point, consider the discretized PDE system:

$$\sum_{\alpha \in A_{\geq m}} \varphi_\alpha^{(\ell)}\left(x_{(i)}, u(x_{(i)}), \partial u(x_{(i)}), \partial^2 u(x_{(i)}), \ldots\right) \partial^\alpha a(x_{(i)})$$
$$= C_{(i)}^{(\ell)}, \quad \ell = 1, \ldots, L.$$

Then, by stacking the equations across all grid points, we

can represent the system as a linear system:

$$\mathbf{M} \cdot \mathbf{a} = \mathbf{c},$$

where $\mathbf{a} \in \mathbb{R}^{|A_{\geq m}| \cdot N}$ is the vector of derivatives of the coefficient function $a(x)$, $\mathbf{M} \in \mathbb{R}^{LN \times |A_{\geq m}| N}$ is the matrix constructed from $\varphi_\alpha^{(\ell)}$, and $\mathbf{c} \in \mathbb{R}^{|A_{\geq m}| \cdot N}$ is the vector of constant values. Then, the following statements are equivalent:

(i) $\text{rank}(\mathbf{M}) = \text{rank}([\mathbf{M}\, \mathbf{c}]) = |A_{\geq m}|N$

(ii) The k-th derivatives of coefficient function $\mathbf{a}$ is uniquely determined, that is, $\mathbf{a}$ has no structural uncertainty.

The proof of Theorem 1 is provided in the Appendix A.

**Remark 1** (Effect of lower-order derivatives). In many PDEs, the coefficient function $a(x)$ enters not only through its $k$-th-order derivatives but also through lower-order derivatives, including the $(k-1)$-st-order derivatives and even the zeroth order $a(x)$ itself. On an infinitesimal grid, and along each coordinate direction, a $(k-1)$-st-order derivative can be viewed as a (discrete) integral of the corresponding $k$-th-order derivative: for example, in one spatial dimension, the value of $a^{(k-1)}$ at a grid point can be written as a cumulative sum of nearby values of $a^{(k)}$, plus an integration constant that corresponds to a lower-degree polynomial term. Thus, at the level of the discretized system, a term in the PDE involving $\partial^{k-1} a$ can, up to such lower-degree polynomial components, be rewritten as a linear combination of $k$-th-order derivatives of $a$.

This observation suggests that lower-order derivative terms are not "wasted" from the viewpoint of our rank-based analysis: after rewriting them in this way, they provide additional linear constraints involving the same $k$-th-order derivatives and may further reduce the effective structural uncertainty. In the present paper, we do not exploit this lifting explicitly and, for clarity, formulate our main theorem solely in terms of the $k$-th-order derivatives that appear directly in the PDE. A systematic extension of our framework that incorporates lifted lower-order terms in multiple spatial dimensions—while carefully accounting for the algebraic relations between mixed partial derivatives and potential path-dependence effects—is technically more complicated and is left for future work. In the example in the next section, we

present a case where the target partial derivative coefficient is uniquely determined by the higher-order terms.

### 3.1. Examples

**• Hamiltonian system**
Let the canonical variables be denoted by $(\boldsymbol{q}, \boldsymbol{p})^\top \in \mathbb{R}^{2d}$, where $\boldsymbol{q} = (q_1, \ldots, q_d)^\top$ are the generalized coordinates and $\boldsymbol{p} = (p_1, \ldots, p_d)^\top$ are the generalized momenta.

Given the Hamiltonian function $H(\boldsymbol{q}, \boldsymbol{p})$, the canonical equations of motion (Hamilton's equations) can be expressed in matrix form as

$$\begin{pmatrix} \frac{d\boldsymbol{q}}{dt} \\ \frac{d\boldsymbol{p}}{dt} \end{pmatrix} =: \begin{pmatrix} \dot{\boldsymbol{q}} \\ \dot{\boldsymbol{p}} \end{pmatrix} = \begin{pmatrix} \mathbf{0} & I_n \\ -I_n & \mathbf{0} \end{pmatrix} \begin{pmatrix} \frac{\partial H}{\partial \boldsymbol{q}} \\ \frac{\partial H}{\partial \boldsymbol{p}} \end{pmatrix}.$$

The equation on an infinitesimal $N$ grid space is written as $\mathbf{M} \cdot \mathbf{a} = \mathbf{c}$, where $M = \begin{pmatrix} \mathbf{0} & I_{Nd} \\ -I_{Nd} & \mathbf{0} \end{pmatrix}$, $\mathbf{a} = \left( \frac{\partial H}{\partial \boldsymbol{q}_1}, \ldots, \frac{\partial H}{\partial \boldsymbol{q}_N}, \frac{\partial H}{\partial \boldsymbol{p}_1}, \ldots, \frac{\partial H}{\partial \boldsymbol{p}_N} \right)^\top$, and $\mathbf{c} = \left( \dot{\boldsymbol{q}}_1, \ldots, \dot{\boldsymbol{q}}_N, \dot{\boldsymbol{p}}_1, \ldots, \dot{\boldsymbol{p}}_N \right)^\top$. Since $\text{rank}(\boldsymbol{M}) = 2Nd$, the necessary conditions are satisfied such that the Hamiltonian function $H(\boldsymbol{q}, \boldsymbol{p})$ is uniquely determined, except for the indefiniteness of the constant. Previous work on HNN has suggested that, when the Hamiltonian is estimated in a data-driven manner, it is identifiable up to a constant (Greydanus et al., 2019b). Also, the structural uncertainty characterized in this example is further examined by numerical experiments in the Appendix B.

**• Lagrange system**
Let the generalized coordinates be denoted by $\boldsymbol{q} = (q_1, \ldots, q_d)^\top$. Lagrange's equations of motion can be written in matrix form as

$$\dot{\boldsymbol{p}} := \frac{d}{dt} \frac{\partial L}{\partial \dot{\boldsymbol{q}}} = \begin{pmatrix} \mathbf{0} & I_d \end{pmatrix} \begin{pmatrix} \frac{\partial L}{\partial \boldsymbol{q}} \\ \frac{\partial L}{\partial \dot{\boldsymbol{q}}} \end{pmatrix}.$$

The equation on an infinitesimal $N$ grid space is written as $\mathbf{M} \cdot \mathbf{a} = \mathbf{c}$, where $\mathbf{M} = \begin{pmatrix} \mathbf{0} & I_{Nd} \end{pmatrix}$, $\mathbf{a} = \left( \frac{\partial L}{\partial \boldsymbol{q}_1}, \ldots, \frac{\partial L}{\partial \boldsymbol{q}_N}, \frac{\partial L}{\partial \dot{\boldsymbol{q}}_1}, \ldots, \frac{\partial L}{\partial \dot{\boldsymbol{q}}_N}, \right)^\top$, and $\mathbf{c} = \left( \dot{\boldsymbol{p}}_1, \ldots, \dot{\boldsymbol{p}}_N \right)^\top$. Since $\text{rank}(\boldsymbol{M}) = Nd < 2Nd$, the Lagrange function $L(\boldsymbol{q}, \dot{\boldsymbol{q}})$ is undetermined.

Since the Hamiltonian and Lagrange systems have a transformable relationship through the Légendre transformation, it seems counterintuitive that only the Lagrangian is not indefinite. The reason the Lagrangian cannot be determined is that the information corresponding to the part of the canonical equation of motion in the Hamiltonian system, $\dot{\boldsymbol{q}} := \frac{\partial H}{\partial \boldsymbol{p}}$, is missing in the Lagrange system. Since one physical constraint for estimating the coefficient function has disappeared, the Lagrange function is not

determined. This missing information corresponds to the definition of the generalized momentum in the Lagrange system, $\boldsymbol{p} := \frac{\partial L}{\partial \dot{\boldsymbol{q}}}$. In fact, adding the definition of generalized momentum to the Lagrangian equation of motion leads to the satisfaction of the necessary condition, $\text{rank}(\boldsymbol{M}) = 2Nd$, for the Lagrangian to be uniquely determined. Also, the structural uncertainty characterized in this example is further examined by numerical experiments in the Appendix C.

**• 1-d diffusion system**
Related to the discussion in Remark 1, we show as an example the case of the one-dimensional diffusion equation

$$\partial_t u(t, x) = \partial_x \big( a(x) \, \partial_x u(t, x) \big), \tag{3}$$

where $a(x)$ is an unknown diffusion coefficient. Here, to discuss the uncertainty when inversely estimating $a(x)$, we consider inversely estimating $\partial_x a(x_j) := \partial_x a(x)|_{x=x_j}$, given at each point $x_j$ on an infinitesimal grid in the $x$-space. Expanding the right-hand side of Eq. (3) gives $\partial_t u(t, x) = \partial_x a(x) \, \partial_x u(t, x) + a(x) \, \partial_{xx} u(t, x)$. From the viewpoint of the coefficient function $a$, this PDE contains the first derivative $\partial_x a(x)$ (order $k - 1 = 1$) with coefficient $\phi(x) = \partial_x u(t, x)$, in addition to the zeroth-order term $a(x)$ with coefficient $\partial_{xx} u(t, x)$.

We discretize the spatial domain on an infinitesimal grid $x_j = x_0 + jh$ $(j = 1, \ldots, N)$ with spacing $h > 0$. On this grid, we denote $s_j := \partial_x a(x_j)$, $t_j := a(x_j)$. Along the one-dimensional grid, the first derivative can be represented as a discrete integral (cumulative sum) of the second derivative:

$$t_j = t_0 + h \sum_{i=1}^{j} s_i, \qquad j = 1, \ldots, N, \tag{4}$$

where $t_0$ is an integration constant corresponding to a lower-degree component of $a(x)$.

Fix a time $t$ and discretize the PDE at the spatial grid points $x_j$: $\partial_t u(t, x_j) = \partial_x a(x_j) \partial_x u(t, x_j) + t_j \partial_{xx} u(t, x_j)$, $j = 1, \ldots, N$. Substituting (4) into this relation yields

$$\partial_t u(t, x_j) = \partial a(x_j) \partial_x u(t, x_j) + \Big( t_0 + h \sum_{i=1}^{j} s_i \Big) \partial_{xx} u(t, x_j).$$

Then, the equation on an infinitesimal $N$ grid space is written as $\mathbf{M} \cdot \mathbf{s} = \mathbf{c}$, where $\mathbf{s} = (s_1, \ldots, s_N)^\top$, $\mathbf{c} = (\partial_t u(t, x_1) - t_0 \partial_{xx} u(t, x_1), \ldots, \partial_t u(t, x_N) - t_0 \partial_{xx} u(t, x_N))$, and

$$M_{ji} = \begin{cases} h \, \partial_{xx} u(t, x_j), & i < j, \\ h \, \partial_{xx} u(t, x_j) + \partial_x u(t, x_j), & i = j, \\ 0, & i > j. \end{cases}$$

If each element of $M$ is nonzero, $M$ is a lower triangular matrix and $\mathrm{rank}(M) = N$. Thus, the diffusion coefficient function $a(x)$ is uniquely determined from the diffusion equation, except for constant term.

Eq. (3) can be regarded as a first-order ordinary differential equation in $a(x)$. Therefore, a first-order is uniquely solvable once a single boundary (or initial) condition is specified, and this single degree of freedom corresponds exactly to an additive constant in $a(x)$. In other words, $\partial_x a(x)$ is uniquely determined. This fact is consistent with the structural uncertainty estimated above.

This simple example illustrates the idea, discussed in Remark 1, that the presence of lower-order partial derivative coefficient terms suggests an effect of reducing the indeterminacy of higher-order partial derivative coefficient functions. The structural uncertainty characterized in this example is further examined by numerical experiments in the Appendix D.

## 4. Proposed Framework

As discussed in Sec. 2, when estimating the coefficient function $a(x)$ using PIML, the structural uncertainty in the underlying physical system may lead to physically inappropriate learning outcomes. Under the mathematical preparation in Sec. 3, we propose the following three-step framework for obtaining scientifically meaningful models within the PIML paradigm.

**Step 1: Diagnose structural uncertainty.**
Before implementing any machine learning model, we evaluate the rank of the matrix $M$ in Theorem 1. This provides information about the degree and structure of the uncertainty of the coefficient function in the given PDE.

**Step 2: Introduce physical constraints.**
Depending on the structure of $M$, we design and incorporate physically induced constraints into the loss function (e.g., Eq. (1) or Eq. (2)) so as to reduce the structural uncertainty. The machine learning model is then trained using this augmented loss.

**Step 3: Analyze dependence on constraint strength.**
We examine how the estimated coefficient function $a(x)$ changes as we vary the strength of the physical constraints. Unlike standard hyperparameter tuning in machine learning, our framework does not select a single "optimal" constraint strength solely by predictive performance. Instead, we provide physicists with the family of solutions obtained over a range of constraint strengths, enabling them to interpret the results in light of domain knowledge. For example, Kepler discovered the law of elliptical orbits by focusing on small systematic discrepancies in planetary motion, rather than on predictive accuracy alone.

## 5. Demonstration

For physics systems that are typically considered in physics-informed machine learning (PIML), including Hamiltonian systems, Lagrange systems, and field-equation-based models, the effectiveness of the proposed framework has already been numerically demonstrated in the Appendix B, C, and D, in correspondence with the Examples discussed in Sec. 3.1. These examples focus on idealized systems commonly studied in PIML and serve to clarify the mechanisms by which structural uncertainty arises.

Building on these results, we here consider a problem setting that is closer to practical physical applications, namely Hamiltonian function estimation based on the wave-kinetic equation. The wave-kinetic equation plays a central role in the kinetic description of classical many-body systems, including nuclear fusion research, and has been theoretically suggested to admit structural uncertainty in coefficient-function estimation, which may not be resolved by data enrichment alone. In this section, we apply the proposed framework to this system in order to explicitly examine the presence and nature of such structural uncertainty in a realistic and challenging setting.

### 5.1. Wave-Kinetic Equation

Modeling the dynamics of turbulent vortices, which emerge in complex, high-dimensional turbulence phenomena observed in fusion reactors, using low-dimensional Hamiltonian dynamical systems, such as the wave-kinetic equation, is useful for the prediction and control of turbulence based on physics understanding (Diamond et al., 2005; Gürcan & Diamond, 2015; Kaw et al., 2001; Sasaki et al., 2017; 2018; Garbet et al., 2021; Sasaki et al., 2021). The wave-kinetic equation describes the time evolution of the density distribution function $I(x, k_x, t)$ in the turbulence phase space $(x, k_x)$ and is given by

$$\partial_t I(z) + \partial_{k_x} H(z)\partial_x I(z) - \partial_x H(z)\partial_{k_x} I(z) = C(z), \quad (5)$$

where $z = (x, k_x, t)$. This equation is mathematically analogous to the Boltzmann equation. Here, the term $C(x, k_x, t)$ represents the generation and damping of turbulent vortices, and is modeled using the linear growth rate $\gamma_L$ and the nonlinear damping rate $\Delta\omega$ as follows: $C(z) := \gamma_L(k_x)I(z) - \Delta\omega[I(z)]^2$, $\gamma_L(k_x) = \frac{k_y(k_x^2 + k_y^2)}{D(1 + k_x^2 + k_y^2)^3} \exp\left(-\left(\frac{k_x}{\Delta k}\right)^2\right)$, where $\Delta k$ characterizes the spectral width of $I(x, k_x, t)$ in the linear regime. The Hamiltonian function $H(x, k_x, t)$, corresponding to the distribution of turbulence intensity, is defined as

$$H(z) = H_0 + \frac{k_y}{1 + k_x^2 + k_y^2} + k_y V_y(x, t). \quad (6)$$

The second term on the right-hand side of Eq. (6) corresponds to the dispersion relation of drift waves, whereas the

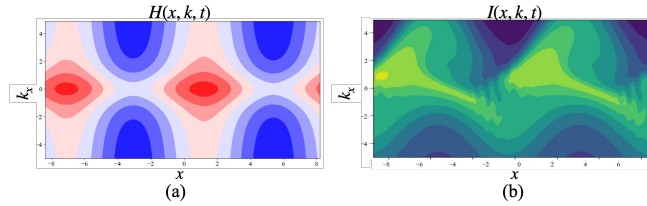

*Figure 2.* (a) Hamiltonian function $H(x, k_x)$ set up in the simulation. (b) Turbulence intensity data $I(x, k_x)$ obtained from the simulation.

third term represents the Doppler shift induced by the zonal flow. In other words, turbulence is deformed (i.e., its spectral distribution is changed) owing to spatially non-uniform Doppler shifts induced by the zonal flow via the third term on the left-hand side of Eq. (5).

Next, we focus on the geodesic acoustic mode (GAM), an oscillatory branch of zonal flows in toroidal plasmas (Dawson et al., 1968). The evolution equation for GAM is given by Sasaki et al. (Sasaki et al., 2018) as

$$
\partial_t^2 V_y(x, t) + \omega_G^2 V_y(x, t)
$$
$$
= \partial_t \partial_x^2 \int dk_x \frac{k_x k_y I(z)}{(1 + k_x^2 + k_y^2)^2} + \mu \partial_t \partial_x^2 V_y(x, t), \quad (7)
$$

where $\omega_G$ is the GAM frequency. The first term on the right-hand side represents the GAM driving term due to Reynolds stress, which is a functional of the turbulent phase space distribution $I(x, k_x, t)$. The turbulence and zonal flows are thus coupled via the third term in Eq. (5) and the first term on the right-hand side of Eq. (7).

In the following analysis, we use numerical solutions of the coupled Eqs. (5) and (7). The parameters used in the simulations are $k_y = 1$, $D = 3$, $\Delta k = 3$, $\omega_G = 0.1061$, and $\mu = 0.05$. This simulation provides the value of $I(x, k_x, t)$ on a grid in the $(x, k_x)$ space. By computing the numerical derivatives from this simulation data, we can obtain the following dataset $DS$ with the sample size $M_x M_{k_x}$: $DS := \left\{ \partial_t I^{(i,j)}, \partial_x I^{(i,j)}, \partial_{k_x} I^{(i,j)} \,\middle|\, i \in [0, M_x], j \in [0, M_{k_x}] \right\}$, where $\partial_{z_k} H^{(i,j)} := \left. \frac{\partial H(z)}{\partial z_k} \right|_{x=x^{(i)}, k=k_x^{(j)}, t=\tau}$, $\partial_{z_k} I^{(i,j)} := \left. \frac{\partial I(z)}{\partial z_k} \right|_{x=x^{(i)}, k=k_x^{(j)}, t=\tau}$, and $C^{(i,j)} := C(x^{(i)}, k_x^{(j)}, \tau)$. Note that in this demonstration, for simplicity, the time slice of the Hamiltonian, $H(x, k_x, t = \tau)$, is estimated independently at each time $\tau$.

The objective of this analysis is to inductively estimate the Hamiltonian function $H(z)$ from the observational data $I(z)$. In understanding the mechanisms of turbulent phenomena based on the coarse-grained wave-kinetic equation, a key bottleneck lies in establishing the correspondence of its simulation results to real-world phenomena. Traditionally, this correspondence is achieved through the manual design of Hamiltonians by scientists based on their insights into physical phenomena. However, designing a

Hamiltonian that accurately reflects complex real-world phenomena—affected by various factors—is generally a challenging task. To assist scientists in the design of such Hamiltonians, we aim to develop a data-driven framework for Hamiltonian estimation. Specifically, we attempt to inversely estimate the Hamiltonian function from measurement data using an HNN-based approach. If successful, this inverse estimation would enable the extraction of physically meaningful information from DNNs and provide valuable support for scientists engaged in Hamiltonian modeling.

**Diagnose structural uncertainty (Step 1)**
Given the turbulence intensity function $I(z)$, we perform an uncertainty evaluation when estimating the Hamiltonian $H(z)$ that the turbulence follows under the constraints of the wave-kinetic equation [Eq. (5)]. First, the wave-kinetic equation is expressed in the infinitesimally small-interval $N_x \times N_{k_x} = \infty \times \infty$ grid space described as matrix form as Eqs. (15) and (16). The size of $\mathbf{M}$ is $N_x N_{k_x} \times 2N_x N_{k_x}$ [see Eq. (16)]. We can see that $\mathrm{rank}(\mathbf{M}) = N_x N_{k_x} < 2N_x N_{k_x}$ and that is why the solution is undefined, and that $N_x N_{k_x}$ of PDEs are not enough to determine the Hamiltonian function uniquely.

**Introduce physical constraints (Step 2)**
The uncertainty of the Hamiltonian estimation is avoided by introducing physical constraints. Assuming now that there is no anisotropy in the $x$ direction in the motion of the system, the Hamiltonian function has line symmetry centered at $k_x = 0$. In fact, the Hamiltonian function used in data generation has line symmetry centered at $k_x = 0$ [Fig. 2(a)]. This constraint implies that $H\left(x^{(i)}, -k_x^{(j)}, t^{(k)}\right) = H\left(x^{(i)}, k_x^{(j)}, t^{(k)}\right)$, $\partial_x H^{(i,-j)} = \partial_x H^{(i,j)}$, and $\partial_{k_x} H^{(i,-j)} = -\partial_{k_x} H^{(i,j)}$, where $\partial_{z_k} H^{(i,-j)} := \left. \frac{\partial H(z)}{\partial z_k} \right|_{x=x^{(i)}, k=-k_x^{(j)}, t=\tau}$. Introducing this constraint into the wave-kinetic equation on the discretized grid reduces the number of independent unknowns, so that $M$ becomes a square matrix $M \in \mathbb{R}^{N_x N_{k_x} \times N_x N_{k_x}}$ [see Eqs (17) and (18) in Appendix]. Since the number of partial differential coefficients of the unknown Hamiltonian is $N_x N_{k_x}$, if $\mathrm{rank}(\mathbf{M}) = N_x N_{k_x}$, the Hamiltonian function is uniquely determined, except for the uncertainty of the constant. For this condition to be satisfied, it must be $\forall i, j$, $\mathrm{rank}\left[ \begin{pmatrix} -\partial_{k_x} I^{(i,j)} & \partial_x I^{(i,j)} \\ -\partial_{k_x} I^{(i,-j)} & -\partial_x I^{(i,-j)} \end{pmatrix} \right] = 2$, where $\partial_{z_k} I^{(i,-j)} := \left. \frac{\partial I(z)}{\partial z_k} \right|_{x=x^{(i)}, k=-k_x^{(j)}, t=\tau}$. This is true if the turbulence intensity distribution $I_x(x, k)$ has a gradient at all points and has no line symmetry centered at $k_x = 0$. Since this is true for the present dataset [Fig. 2(b)], the Hamiltonian function is physically uniquely determined by adding the symmetry constraint, except for the uncertainty of the constant. According to the results of the above evaluation of uncertainty, we designed the loss function as

$$
\mathrm{Loss}(\theta_{\mathrm{nn}}) =
$$
$$
\mathbb{E}\left[ \left\| \partial_t I^{(i,j)} - C^{(i,j)} - \partial_{k_x} H_{\theta_{\mathrm{nn}}}^{(i,j)} \partial_x I^{(i,j)} + \partial_x H_{\theta_{\mathrm{nn}}}^{(i,j)} \partial_{k_x} I^{(i,j)} \right\|^2 \right]
$$
$$
+ \lambda \mathbb{E}\left[ \left\| H_{\theta_{\mathrm{nn}}}(x_i, k_j) - H_{\theta_{\mathrm{nn}}}(x_i, -k_j) \right\|^2 \right], \quad (8)
$$

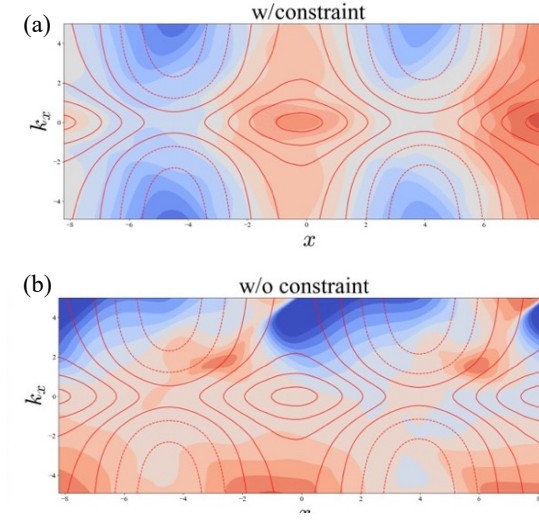

*Figure 3.* (a) Estimation results for the Hamiltonian function $H_{\theta_{nn}}(x, k_x)$ with symmetry constraints and (b) without constraints. The histogram represents the DNN function estimation results, and the red contour line represents the true Hamiltonian function set when generating the dataset.

where $\mathbb{E}$ is the averaging operator with respect to $i$ and $j$. For further details on the neural network model and other aspects, please refer to Appendix E.

**Analyze dependence on constraint strength $\lambda$ (Step 3)**
Training was performed using the loss function in Eq. (8). Please refer to the supplemental material for details on the parameters used in the training. The estimation results for the constrained and unconstrained cases are shown in Figs. 3(a) and 3(b), respectively. Also, please refer to Appendix G, which contains video information regarding the estimation results of the Hamiltonian time series. As shown in the results, the introduction of symmetry constraints enabled the neural-network-modeled Hamiltonian function (heat map) to capture the features of the original Hamiltonian function (red contour lines) set at the time of dataset generation. In the case of without constraints, a Hamiltonian significantly deviating from the original Hamiltonian was learned. This result was confirmed not only through visual comparison but also through the quantitative comparison of the similarity between the true Hamiltonian and the Hamiltonian estimated by the DNN [Table 2]. As stated in Theorem 1, the Hamiltonian retains uncertainty in the (k-1)-polynomial order term, i.e., the constant term. Therefore, note that when calculating the similarity of the Hamiltonian, PIML corrects the estimated Hamiltonian values so that their average matches the average of the true Hamiltonian. The values of the loss function for the validation data revealed that the unconstrained case had better prediction performance than the constrained case [Table 2]. This comparison corresponds to

the search for the hyperparameter $\lambda$ associated with **step 3** . The reversal of the Hamiltonian estimation accuracy and prediction performance indicates the danger of determining the hyperparameters based on the prediction performance, as described in **step 3**.

*Table 2.* Cosine similarity between the Hamiltonian function estimated by DNN and the true Hamiltonian function set when generating the dataset, and mean prediction error [the first term of the loss function Eq. (8)] for the validation data. Mean $\pm$ standard error of cosine similarity and prediction errors over 30 independent trials. Note that a higher value is better for cosine similarity.

| constraint | cosine similarity | L2-loss |
|---|---|---|
| W/ | **0.45** $\pm$ 0.29 | $(5.46 \pm 0.48) \times 10^{-8}$ |
| W/O | 0.06 $\pm$ 0.09 | $(\textbf{2.58} \pm 0.27) \times 10^{-8}$ |

## 6. Discussion

This study addressed structural uncertainty in PIML, focusing on the non-uniqueness inherent in the data-driven estimation of PDE coefficient functions. By introducing a rank-based criterion, we demonstrated that such uncertainty can be quantitatively characterized at the level of the physical model itself, independently of data noise or model expressiveness.

Our analysis highlights that structural uncertainty is fundamentally different from model-form and data uncertainty. While the latter can be mitigated by expanding the model class or incorporating statistical treatments, structural uncertainty originates from intrinsic properties of the governing equations, such as symmetry or gauge invariance, and therefore requires explicit physical constraints to enforce identifiability. This distinction clarifies why minimizing predictive loss alone may lead to physically inconsistent solutions in PIML.

Through the wave-kinetic Hamiltonian learning example, we showed that unconstrained optimization yields multiple coefficient functions with comparable predictive performance, whereas incorporating physically motivated constraints resolves non-uniqueness and enables physically consistent identification. These results emphasize the importance of diagnosing structural uncertainty prior to tuning constraints based solely on prediction accuracy.

We note that the theoretical framework developed in this study is restricted to partial differential equations that are linear with respect to the coefficient functions. Nevertheless, many physics models are formulated in a form that is linear in their coefficients, and this assumption encompasses a broad and practically important class of physical systems. Extending the proposed criterion to coefficient-nonlinear PDEs remains an important direction for future research.

IMPACT STATEMENT

This paper presents work whose goal is to advance the field of Machine Learning and science. There are many potential societal consequences of our work, none which we feel must be specifically highlighted here.

ACKNOWLEDGMENTS

Figure 1 was generated using DALL-E3, OpenAI.

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

## A. Proof of Theorem 1

We prove the equivalence of (i) and (ii) in Theorem 1.

**Proof of (i) ⇒ (ii):**
If $\text{rank}(\mathbf{M}) = \text{rank}(\mathbf{M}, \mathbf{c}) = |A_{\geq m}|N$, the $k$-leaf set of derivatives $S_k^{\text{leaf}}$ is determined on arbitrarily positioned $x$. In other words, the $k$-th-order partial differential coefficients of $a(x)$ are uniquely determined on an infinitesimally small-spaced grid. Then, it is shown that the partial differential coefficient function $a(x)$ is uniquely determined except for the uncertainty of the $k - 1$ degree polynomial as follows. Consider the case $k = 0$, $d = 2$. If $x = (x, y)$, then

$$a(x,y) - a(x_0, y_0) = [a(x, y_0) - a(x_0, y_0)] + [a(x, y) - a(x, y_0)]$$

Applying the fundamental theorem of calculus to the right-hand side, we obtain

$$a(x,y) - a(x_0, y_0) = \int_{x_0}^{x} dx \partial_x a(x, y_0) + \int_{y_0}^{y} dy \partial_y a(x, y).$$

Furthermore, from the definition of integral, it can be transformed as follows.

$$a(x,y) - a(x_0, k_0) = \lim_{\Delta_x \to 0} \sum_{i=1}^{n_x} \partial_x a(x_0 + i\Delta_x, y_0)\Delta_x$$
$$+ \lim_{\Delta_y \to 0} \sum_{j=1}^{n_k} \partial_y a(x, y_0 + j\Delta_k)\Delta_y$$

Because the $k$-th-order partial differential coefficients of $a(x)$ are uniquely determined on an infinitesimal small spaced grid, the right-hand side can be calculated. Thus, it is shown that the partial differential coefficient function $a(x,y)$ at arbitrary coordinates $(x, y)$ can be uniquely estimated except for the uncertainty of the constant $a(x_0, k_0)$.

For general $k$ and $d$, we also decompose the expression as

$$\partial^k a(x_1, x_2, \ldots, x_d) - \partial^k a(x_1^{(0)}, x_2^{(0)}, \ldots, x_d^{(0)}) =$$
$$\left[\partial^{k+1} a(x_1, x_2^{(0)}, \ldots, x_d^{(0)}) - \partial^{k+1} a(x_1^{(0)}, x_2^{(0)}, \ldots, x_d^{(0)})\right] +$$
$$\cdots \left[\partial^{k+1} a(x_1, \ldots, x_{d-1}, x_d) - \partial^{k+1} a(x_1, \ldots, x_{d-1}, x_d^{(0)})\right].$$

Then, each term transforms to an integral form in the same manner as the case of $k = 0, d = 2$, completing the inductive argument.

**Proof of (ii) ⇒ (i):**
Assume that $\mathbf{M} \cdot \mathbf{a} = \mathbf{c}$ has a unique solution $\mathbf{a}^\star$. Let $\mathbf{x} \in \text{ker}(\mathbf{M})$ be arbitrary, so that $\mathbf{Mx} = \mathbf{0}$. Then

$$\mathbf{M}(a^\star + \mathbf{x}) = \mathbf{Ma}^\star + \mathbf{Mx} = \mathbf{c} + \mathbf{0} = \mathbf{c},$$

which implies that $\mathbf{a}^\star + \mathbf{x}$ is also a solution of $\mathbf{Ma} = \mathbf{c}$. By uniqueness of the solution, we must have $\mathbf{a}^\star + \mathbf{x} = \mathbf{a}^\star$, and hence $\mathbf{x} = \mathbf{0}$. Therefore,

$$\text{ker}(\mathbf{M}) = \{0\}.$$

A linear map $\mathbf{M}$ has a trivial kernel if and only if it is injective, which is equivalent to its columns being linearly independent. Thus,

$$\text{rank}(\mathbf{M}) = n,$$

and $\mathbf{M}$ has full rank. ∎

Note that if $\text{rank}(\mathbf{M}) \neq |A_{\leq m}|N$, but the $k$-leaf set of derivatives is uniquely determined, then the coefficient function $a(x)$ is also uniquely determined up to a polynomial of degree of at most $k-1$. In this case, the proofs and proposed methods can be set up in the same manner. Also, even when some $k$th-order partial differential coefficients are undefined or not included in the PDEs, the same argument holds if the corresponding lower-order partial differential coefficients can be estimated.

## B. Demonstration in a Hamiltonian System

As discussed in Sec. 3.1, when estimating a Hamiltonian function from the canonical equations of motion, structural uncertainty does not arise provided that the associated matrix $M$ is $\text{rank}(M) = N$; it emerges only for datasets that induce a rank deficiency in $M$. In terms of our formulation, this corresponds to the case where the associated linear system satisfies $\text{rank}(M) = N$.

We verify this property through a Hamiltonian estimation task for a simple pendulum system. The ground-truth Hamiltonian is given by

$$H_{\text{true}}(q, p) = \frac{1}{2}ml^2p^2 + mgl\left(1 - \cos q\right), \qquad (9)$$

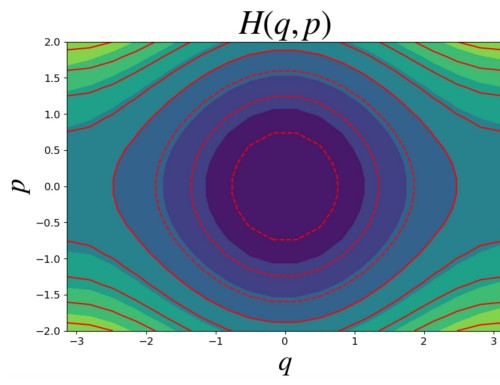

*Figure 4.* Estimation results for the Hamiltonian function $H_{\boldsymbol{\theta}_{\mathrm{nn}}}(q, p)$. The histogram shows the distribution of the estimated Hamiltonian values obtained by the DNN, while the red contour indicates the ground-truth Hamiltonian level sets used to generate the dataset.

where $q := \theta$, $p := \dot{\theta}$, and the parameters are fixed to $m = 1.0$, $g = 1.0$, and $l = 1.0$.

In this demonstration, we consider the problem of estimating the Hamiltonian function $H(q, p)$ from observational data, consisting solely of time-series measurements of the state variables $(q, p)$. A Hamiltonian neural network $H_{\boldsymbol{\theta}_{\mathrm{nn}}}$ is trained to approximate the true Hamiltonian.

The loss function used to optimize the network parameters $\boldsymbol{\theta}_{\mathrm{nn}}$ is defined as

$$
\begin{aligned}
\mathrm{Loss}(\boldsymbol{\theta}_{\mathrm{nn}}) &= \frac{1}{N} \left[ \sum_i \|\partial_p H_{\boldsymbol{\theta}_{\mathrm{nn}}}(q_i, p_i) - \dot{q}_i\|^2 \right. \\
&+ \left. \|\partial_q H_{\boldsymbol{\theta}_{\mathrm{nn}}}(q_i, p_i) + \dot{p}_i\|^2 \right].
\end{aligned}
$$

As shown in Fig. 4, the trained Hamiltonian $H_{\boldsymbol{\theta}_{\mathrm{nn}}}$ converges to a unique solution that is consistent with the ground truth, exhibiting no structural uncertainty. This result serves as a baseline example and confirms that the proposed uncertainty assessment framework correctly identifies cases in which the Hamiltonian function is uniquely determined by the observed dynamics.

## C. Demonstration in a Lagrange system

As discussed in Sec. 3.1, when estimating a Lagrangian function from the Euler-Lagrange equations, structural uncertainty generally arises irrespective of the dataset provided. We verify this property through a Lagrangian estimation task for a simple pendulum system. The ground-truth Lagrangian is given by

$$
L_{\mathrm{true}}(\theta, \dot{\theta}) = \frac{1}{2} m l^2 \dot{\theta}^2 - mgl\left(1 - \cos\theta\right), \quad (10)
$$

where the parameters are fixed to $m = 1.0$, $g = 1.0$, and $l = 1.0$.

In this demonstration, we consider the problem of estimating the Lagrangian function $L(\theta, \dot{\theta})$ from observational data consisting solely of time-series measurements of the state variables $(\theta, \dot{\theta})$. A Lagrangian neural network $L_{\boldsymbol{\theta}_{\mathrm{nn}}}$ is trained to approximate the true Lagrangian.

We first consider an unconstrained setting. The loss function used to optimize the network parameters $\boldsymbol{\theta}_{\mathrm{nn}}$ is defined as

$$
\mathrm{Loss}^{\mathrm{w/o}}(\boldsymbol{\theta}_{\mathrm{nn}}) = \frac{1}{N} \sum_i \left\| \partial_\theta L_{\boldsymbol{\theta}_{\mathrm{nn}}}(\theta_i, \dot{\theta}_i) - \ddot{\theta}_i \right\|^2. \quad (11)
$$

As discussed in Sec. 3.1, the structural uncertainty inherent in the Lagrangian formulation can be reduced by introducing additional constraints derived from the generalized momentum, $\boldsymbol{p} := \partial L / \partial \dot{\boldsymbol{q}}$. We next demonstrate Lagrangian estimation under this constrained setting.

The loss function used to optimize the network parameters $\boldsymbol{\theta}_{\mathrm{nn}}$ with the momentum constraint is defined as

$$
\begin{aligned}
\mathrm{Loss}^{\mathrm{w/}}(\boldsymbol{\theta}_{\mathrm{nn}}) &= \frac{1}{N} \sum_i \left[ \left\| \partial_\theta L_{\boldsymbol{\theta}_{\mathrm{nn}}}(\theta_i, \dot{\theta}_i) - \ddot{\theta}_i \right\|^2 \right. \\
&+ \left. \lambda \left\| \partial_{\dot{\theta}} L_{\boldsymbol{\theta}_{\mathrm{nn}}}(\theta_i, \dot{\theta}_i) - \dot{\theta}_i \right\|^2 \right], \quad (12)
\end{aligned}
$$

where $\lambda = 1$. As shown in Fig. 5(a), the trained Lagrangian $L_{\boldsymbol{\theta}_{\mathrm{nn}}}$ under the momentum constraint converges to a unique solution consistent with the ground truth, exhibiting no structural uncertainty. In contrast, when the constraint is not imposed, multiple functionally distinct Lagrangians are consistent with the observed dynamics, resulting in pronounced structural uncertainty [Fig. 5(b)].

Table 3 provides a quantitative comparison of these results. Notably, the regression performance, measured by the prediction loss, remains nearly unchanged with or without the constraint. This highlights that predictive accuracy alone is insufficient to diagnose structural uncertainty. Together, these results demonstrate that the proposed uncertainty assessment framework successfully distinguishes between identifiable and non-identifiable Lagrangian estimation settings.

*Table 3.* Cosine similarity between the Lagrangian function estimated by DNN and the true Lagrangian function set when generating the data, and mean value of the loss function [Eq. (11)] for the dataset. Mean $\pm$ standard error of cosine similarity and prediction errors over 5 independent trials with different dataset. Note that a higher value is better for cosine similarity; a lower value is better for L2-loss.

| constraint | cosine similarity | L2-loss |
|:---:|:---:|:---:|
| W/ | **1.00** $\pm$ 0.00 | $(7.76 \pm 1.97) \times 10^{-2}$ |
| W/O | 0.60 $\pm$ 0.10 | $(7.61 \pm 1.92) \times 10^{-2}$ |

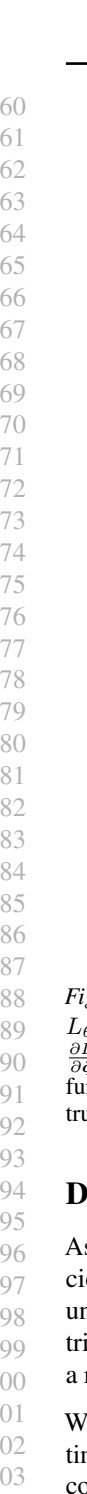

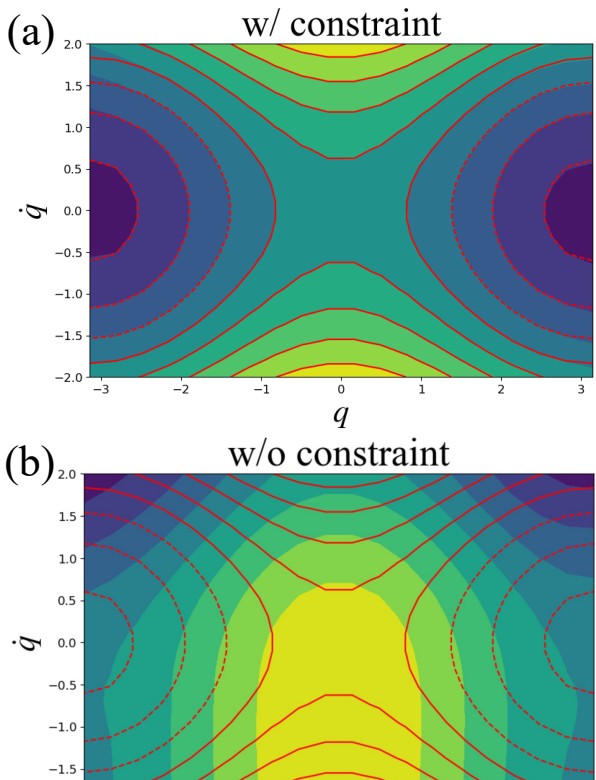

(a) w/ constraint

(b) w/o constraint

*Figure 5.* (a) Estimation results for the Lagrangian function $L_{\theta_{\mathrm{nn}}}(\theta, \dot{\theta})$ with constraints of the generalized momentum $\boldsymbol{p} := \frac{\partial L}{\partial \dot{q}}$ and (b) without constraints. The histogram represents the DNN function estimation results, and the red contour line represents the true Lagrangian function set when generating the dataset.

## D. Demonstration in a 1-d diffusion system

As discussed in Sec. 3.1, when estimating a diffusion coefficient function from the 1-d diffusion equations, structural uncertainty does not arise provided that the associated matrix $M$ is full rank; it emerges only for datasets that induce a rank deficiency in $\mathrm{rank}(M) = N$.

We verify this property through a numerical experiments estimating diffusion-coefficient. The ground-truth of diffusion coefficient function is given by

$$a_{\mathrm{true}}(x) = 0.2 + 0.15(1.0 + \sin(k\pi x))\exp(-3.0x), \quad (13)$$

where $k$ is set as $\{1, 2, 3, 4, 5\}$ to generate five datasets. To generate datasets, we set Dirichlet boundary conditions at t = 0 as follows:

$$u(x, 0) = \sin(\pi x).$$

In this demonstration, we consider the problem of estimating the diffusion coefficient function $a(x)$ from observational

data $u(x, t)$, consisting solely of time-series measurements of the state variables $(t, x)$. A Diffusion coefficient neural network $a_{\boldsymbol{\theta}_{\mathrm{nn}}}$ is trained to approximate the true Hamiltonian.

The loss function used to optimize the network parameters $\boldsymbol{\theta}_{\mathrm{nn}}$ is defined as

$$\mathrm{Loss}(\boldsymbol{\theta}_{\mathrm{nn}}) = \frac{1}{N}\left[\sum_i \|\partial_t u(t_i, x_i)\right.$$
$$\left. - \{\partial_x a(x_i)\, \partial_x u(t_i, x_i) + a(x_i)\, \partial_{xx} u(t_i, x_i)\}\|^2\right].$$

As shown in Fig. 6(a), the trained diffusion coefficient function $a_{\boldsymbol{\theta}_{\mathrm{nn}}}$ converges to a unique solution that is consistent with the ground truth, exhibiting no structural uncertainty. We further verified our method on datasets where the rank of matrix M decreases. When the Dirichlet boundary conditions are set as follows,

$$u(x, 0) = 0,$$

$u(x, t)$ becomes zero throughout the entire domain, and consequently its partial derivatives also become zero. As a result, the rank of matrix $M$ becomes zero (see Sec. 3.1). It has been confirmed that the diffusion coefficient function becomes uncertain in such datasets [Fig. 6(b)].

Table 4 provides a quantitative comparison of these results. Notably, the regression performance, measured by the prediction loss, remains nearly unchanged $\mathrm{rank}(M) = N$ or $\mathrm{rank}(M) = 0$ the constraint. This highlights that predictive accuracy alone is insufficient to diagnose structural uncertainty. Together, these results demonstrate that the proposed uncertainty assessment framework successfully distinguishes between identifiable and non-identifiable Lagrangian estimation settings.

## E. DNN Model and its Training Parameters

Here, we describe the DNN models and their training settings. In this study, we used a fully coupled multilayer neural network as the DNN model. The DNNs consisted of an input layer, two hidden layers, and an output layer. The number of nodes in each layer was set as shown in the

*Table 4.* Cosine similarity between the diffusion coefficient function estimated by DNN and the true diffusion coefficient set when generating the data, and mean value of the loss function for the validation data. Mean ± standard error of cosine similarity and prediction errors over 5 independent trials. Note that a higher value is better for cosine similarity; a lower value is better for L2-loss.

| rank($M$) | cosine similarity | L2-loss |
|---|---|---|
| $N$ | **1.00** ± 0.00 | $(1.67 \pm 1.65) \times 10^{-4}$ |
| 0 | 0.17 ± 0.87 | 0.0 ± 0.0 |

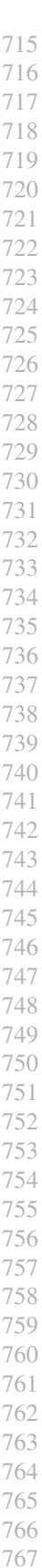

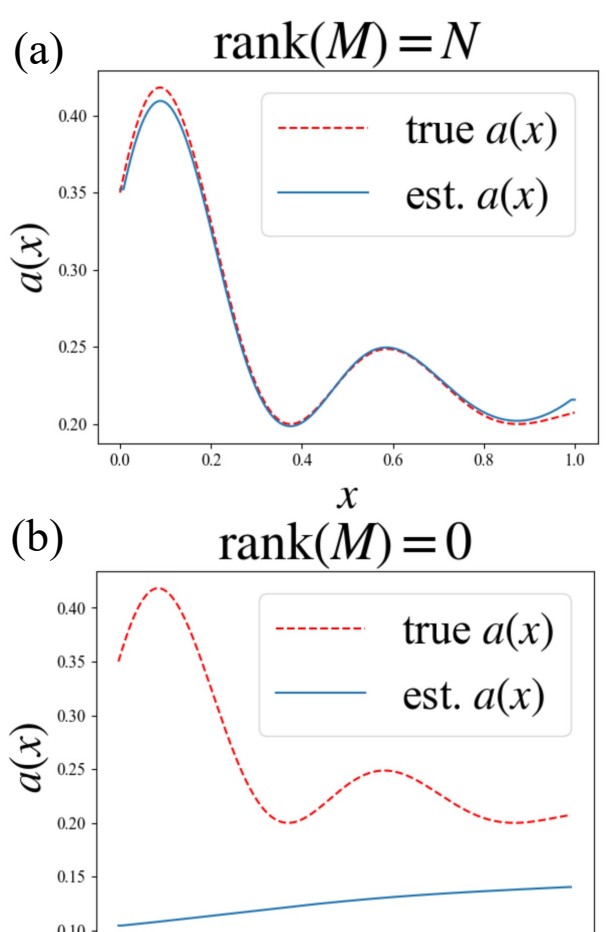

(a) rank$(M) = N$

(b) rank$(M) = 0$

*Figure 6.* (a) Estimation results for the diffusion coefficient function $a(x)$ for a dataset with rank$(M) = N$, and (b) for a dataset with rank$(M) = 0$. The histogram shows the distribution of the diffusion coefficient estimates obtained by the DNN, while the red curve indicates the ground-truth diffusion coefficient used to generate the dataset.

"Network structure" in Table 5. The activation functions of the DNNs were set as the hyperbolic tangent function as shown in the "Activation function" in Table 5. The tanh function is defined as

$$\tanh(x) = \frac{\exp(x) - \exp(-x)}{\exp(x) + \exp(-x)}. \quad (14)$$

The number of samples used for training the DNN model is shown in Table 5 as "Training data size $N$". The Adam method (Kingma & Ba, 2014) was used for training. The training iterations are shown in Table 5. For other details, please see the code shared at https://anonymous.4open.science/r/Structural_uncertainty-30D5.

*Table 5.* Parameters of DNN model and its training. In the "Network structure", the number of nodes is shown in the order from left to right: input layer – first layer – second layer – third layer – output layer.

| Parameter name | Case | |
|---|---|---|
| Training data size $N$ | Hamiltonian | 50,000 |
| | Lagrangian | 50,000 |
| | Diffusion | 160,801 |
| | wave-kinetic | 10,000 |
| Activation function | all | tanh |
| Training algorithm | all | Adam |
| Network structure | Hamiltonian | 2-64-64-1 |
| | Lagrangian | 2-64-64-1 |
| | Diffusion | 1-64-64-1 |
| | wave-kinetic | 2-100-10-1 |
| Training iteration | Hamiltonian | 2,000 |
| | Lagrangian | 2,000 |
| | Diffusion | 4,000 |
| | wave-kinetic | 400,000 |

## F. Details of the uncertainty evaluation in wave-kinetic theory

This section describes the uncertainty assessment in Step 1 and the uncertainty assessment after introducing physical constraints in Step 2. Please refer to Sec. 5.1 for the definitions of each variable.

**Diagnose structural uncertainty (Step 1)**
Given the turbulence intensity function $I(z)$, we perform an uncertainty evaluation when estimating the Hamiltonian $H(z)$ that the turbulence follows under the constraints of the wave-kinetic equation [Eq. (5)]. First, the wave-kinetic equation is expressed in the infinitesimally small-interval $N_x \times N_{k_x} = \infty \times \infty$ grid space as follows:

$$\begin{pmatrix} \partial_t I^{(1,1)} - C^{(1,1)} \\ \partial_t I^{(1,2)} - C^{(1,2)} \\ \vdots \\ \partial_t I^{(N_x, N_{k_x}-1)} - C^{(N_x, N_{k_x}-1)} \\ \partial_t I^{(N_x, N_{k_x})} - C^{(N_x, N_{k_x})} \end{pmatrix} = \mathbf{M} \begin{pmatrix} \partial_x H^{(1,1)} \\ \partial_{k_x} H^{(1,1)} \\ \vdots \\ \partial_x H^{(N_x, N_{k_x})} \\ \partial_{k_x} H^{(N_x, N_{k_x})} \end{pmatrix},$$

(15)

$$\mathbf{M} := \begin{pmatrix} -\partial_{k_x} I^{(1,1)} & \partial_x I^{(1,1)} & 0 & 0 \\ 0 & 0 & -\partial_{k_x} I^{(1,2)} & \partial_x I^{(1,2)} \\ & & & & \ddots \end{pmatrix},$$

(16)

where the size of $\mathbf{M}$ is $N_x N_{k_x} \times 2N_x N_{k_x}$. We can see that rank$(\mathbf{M}) = N_x N_{k_x} < 2N_x N_{k_x}$ and that is why the solution is undefined, and that $N_x N_{k_x}$ of PDEs are not

enough to determine the Hamiltonian function uniquely.

**Introduce Physical Constraints (Step 2)**

The uncertainty of the Hamiltonian estimation is avoided by introducing physical constraints. Assuming now that there is no anisotropy in the $x$ direction in the motion of the system, the Hamiltonian function has line symmetry centered at $k_x = 0$. In fact, the Hamiltonian function used in data generation has line symmetry centered at $k_x = 0$ [Fig. 2(a)]. This constraint implies that $H\left(x^{(i)}, -k_x^{(j)}, t^{(k)}\right) = H\left(x^{(i)}, k_x^{(j)}, t^{(k)}\right)$, $\partial_x H^{(i,-j)} = \partial_x H^{(i,j)}$, and $\partial_{k_x} H^{(i,-j)} = -\partial_{k_x} H^{(i,j)}$, where $\partial_{z_k} H^{(i,-j)} := \left. \frac{\partial H(z)}{\partial z_k} \right|_{x=x^{(i)}, k=-k_x^{(j)}, t=\tau}$. Introducing this constraint into the wave-kinetic equation on the grids gives the following representation with block matrices.

$$
\begin{pmatrix}
\partial_t I^{(1,1)} - C^{(1,1)} \\
\partial_t I^{(1,-1)} - C^{(1,-1)} \\
\vdots \\
\partial_t I^{(N_x, \frac{N_{k_x}}{2})} - C^{(N_x, \frac{N_{k_x}}{2})} \\
\partial_t I^{(N_x, \frac{-N_{k_x}}{2})} - C^{(N_x, \frac{-N_{k_x}}{2})}
\end{pmatrix}
= \boldsymbol{M}
\begin{pmatrix}
\partial_x H x^{(1,1)} \\
\partial_{k_x} H^{(1,1)} \\
\vdots \\
\partial_x H^{(\frac{N_x}{2}, \frac{N_{k_x}}{2})} \\
\partial_{k_x} H^{(\frac{N_x}{2}, \frac{N_{k_x}}{2})}
\end{pmatrix}, \ (17)
$$

$$
\boldsymbol{M} =
\begin{pmatrix}
-\partial_{k_x} I^{(1,1)} & \partial_x I^{(1,1)} & & & \\
-\partial_{k_x} I^{(1,-1)} & -\partial_x I^{(1,-1)} & & & \boldsymbol{0} \\
& & \ddots & & \\
& \boldsymbol{0} & & -\partial_{k_x} I^{(\frac{N_x}{2}, \frac{N_{k_x}}{2})} & \partial_x I^{(\frac{N_x}{2}, \frac{N_{k_x}}{2})} \\
& & & -\partial_{k_x} I^{(\frac{N_x}{2}, \frac{-N_{k_x}}{2})} & -\partial_x I^{(\frac{N_x}{2}, \frac{-N_{k_x}}{2})}
\end{pmatrix},
$$

$$(18)$$

where $\partial_{z_k} I^{(i,-j)} := \left. \frac{\partial I(z)}{\partial z_k} \right|_{x=x^{(i)}, k=-k_x^{(j)}, t=\tau}$, $C^{(i,-j)} := C(x^{(i)}, -k_x^{(j)}, \tau)$, and the size of $\mathbf{M}$ is $N_x N_{k_x} \times N_x N_{k_x}$. Since the number of partial differential coefficients of the unknown Hamiltonian is $N_x N_{k_x}$, if $\mathrm{rank}(\mathbf{M}) = N_x N_{k_x}$, the Hamiltonian function is uniquely determined, except for the uncertainty of the constant. For this condition to be satisfied, it must be $\forall i, j, \ \mathrm{rank}\left[\begin{pmatrix} -\partial_{k_x} I^{(i,j)} & \partial_x I^{(i,j)} \\ -\partial_{k_x} I^{(i,-j)} & -\partial_x I^{(i,-j)} \end{pmatrix}\right] = 2$. This is true if the turbulence intensity distribution $I_x(x, k)$ has a gradient at all points and has no line symmetry centered at $k_x = 0$. Since this is true for the present dataset [Fig. 2(b)], the Hamiltonian function is physically uniquely determined by adding the symmetry constraint, except for the uncertainty of the constant. According to the results of the above evaluation of uncertainty, we designed the loss function as

$$
\mathrm{Loss}(\theta_{\mathrm{nn}}) =
$$

$$
\mathbb{E}\left[\left\| \partial_t I^{(i,j)} - C^{(i,j)} - \partial_{k_x} H_{\theta_{\mathrm{nn}}}^{(i,j)} \partial_x I^{(i,j)} + \partial_x H_{\theta_{\mathrm{nn}}}^{(i,j)} \partial_{k_x} I^{(i,j)} \right\|^2\right]
$$

$$
+ \lambda \mathbb{E}\left[\left\| H_{\theta_{\mathrm{nn}}}(x_i, k_j) - H_{\theta_{\mathrm{nn}}}(x_i, -k_j) \right\|^2\right], \quad (19)
$$

where $\mathbb{E}$ is the averaging operator with respect to $i$ and $j$. For further details on the neural network model and other aspects, please refer to Appendix E.

## G. Video of Hamiltonian Estimation Results

Please refer to the attached files of "movie.gif", with symmetry constraints, and "movie_withoutconst.gif", without symmetry constraints, at https://anonymous.4open. science/r/Structural_uncertainty-30D5. The files show all the estimated Hamiltonian functions at each time as movies.

