# OpenReview forum: "Uncertainties in Physics-informed Inverse Problems: The Hidden Risk in Scientific AI"
_ICML.cc/2026/Conference — Submitted to ICML 2026_

### Official Review · Reviewer_H9rF · 2026-03-11

**Soundness:** 3
**Presentation:** 3
**Significance:** 3
**Originality:** 3
**Overall Recommendation:** 4
**Confidence:** 4

**Summary:**

Physics Informed Inverse Problems can be learnt when PDE is integrated into machine learning models by Optimizing predictive performance. Mostly in this domain, we find out coefficient functions that are learnt from the observational data. This approach can be significantly affected by Structural Uncertainty, Model form uncertainty and Data form uncertainty. This paper provides framework that quantify and analyze structural uncertainty in the estimation of PDE coefficient functions within Physics Informed Machine Learning(PIML). Framework shows how integrating geometric constraints can lead to unique identification in comparison to purely predictive learning. Study is focused on partial differential equations that are linear with respect to the coefficient function and requires further more validation on different complex PDEs.

**Compliance With Llm Reviewing Policy:**

Affirmed.

**Final Justification:**

Paper is technically sound and  This paper provides framework that quantify and analyze structural uncertainty in the estimation of PDE coefficient functions within Physics Informed Machine Learning(PIML). Initial concern was limited literature survey  and majorly I had questions about related work discussion focused on and before paper published in 2022, I wanted to know from authors if limited survey was done or they claim that not much work has been done in recent years. In Rebuttal authors provided explanation for this along with few references and they claim that structural uncertainty is largely unexplored.

Regarding Originality , Framework provides researcher with the family of solutions obtained not just one optimal solution which then will help researcher to chose the right one instead of accepting the single output. Scope of experimentation is limited and I had question about the generalizability for which authors responded with few more examples and mentioned that they will add it in revised version.

Authors answered my major Questions and hence I decided to increase score by 1 , but for future work they need to work on generalizability more but given the idea of exploring structural uncertainty, good presentation of work and novel theoretical framework I think this can be good contribution to field.

**Key Questions For Authors:**

1) Any specific reason of not including recent works done in year 2023,2024 and 2025. Are you claiming that very limited amount of work is done in this domain in last three years and whatever work is done , that has been included in the paper. I understand that this is mathematical framework but even there are some agentic systems that work in this direction, discussion related to them in the paper would have made it more comprehensive. I would recommend to revise related work section.

2) There is much repetition about types of uncertainty, It is evident from the paper that structural uncertainty is being addressed but mentioning about all three uncertainty in introduction and then again in section 2, seems like elongation of paper. Please write it in one section and in concise manner.

3) Regarding the Mathematical framework, I would appreciate the way it is described in the paper but current experimentation scope seems limited. How can we generalize the framework at this initial stage. Please clarify.

I feel authors should spend some more time on reading recent work and perform more experiments, compare with baselines.

**Limitations:**

Yes

**Strengths And Weaknesses:**

Here is thorough Assessment of Strength and Weakness :

1) Soundness : Submission is technically sound. Paper has significant amount of theoretical analysis and proper proofs and derivation along with detailed mathematical explanation is provided. Claims are well supported and Appendix has all other required proofs and derivations which couldn't fit in the main paper. Authors have mentioned about the few limitations in the paper and which they will address in future. Potentially there are few more limitations than discussed in the paper. Current study involves partial differential equations that are linear with respect to the coefficient functions but no discussion on Ordinary Differential Equations is done.

2) Presentation : Paper is well written and structured. It involves heavy mathematics but language kept is simple to understand complex mathematical equations as well. Narrative is easy to follow only if reader had some prior knowledge about Linear Algebra and Calculus. Major concern is regarding the discussion of prior work, Paper majorly cites paper around year 2019,2021,2022 and there are only 2 papers cited from 2023, 2024 and 1 paper in 2025. Even they claim PIML is emerging field. I agree but the development in this field is so rapid that we have already moved to AI agents to solve many equations discovery in PIML. It seems like recent literature is not referred. Theoretical frameworks may be less but atleast discussion around the uncertainty quantification in recent years should have been discussed and then they could have positioned themselves.

3) Significance : Paper clearly states that it focus on addressing structure uncertainty and relevant experimentation and theoretical analysis is done on the same. Currently study seems limited and majorly focused on Hamiltonian functions. It can help researchers in future but  authors need to do through literature survey and then make some modifications.

4) Originality : The theoretical framework is strong and well articulated. Framework provides researcher with the family of solutions obtained not just one optimal solution. This will help researchers to understand and validate the output. They are diagnosing the structural uncertainty before implementing the ML Model with the mathematical framework and would make outputs more reliable.

---

> ### Author Rebuttal · Authors · 2026-03-30
>
> We sincerely thank the reviewer for the careful and constructive feedback, and for recognizing the clarity and theoretical soundness of our framework. We address the key concerns below.
>
>
> **[1. Recent literature]**
>
> We appreciate the reviewer’s comment regarding recent work (2023–2025). Our statement about limited prior work specifically refers to structural uncertainty in PDE coefficient functions, which, to our knowledge, has very limited direct prior work due to the intrinsic mathematical difficulty of PDEs. In contrast, structural identifiability of scalar parameters in ODE and compartment-type models has been extensively studied. We note that these relevant studies were not sufficiently covered in the original submission, and we will explicitly include representative works such as Ljung & Glad (1994); Miao et al. (2011); Meshkat & Sullivant (2014); Meshkat et al. (2018). We will clarify that our contribution is closely related to this line of research, while addressing the substantially different setting of function-valued coefficients in PDE-based physics-informed inverse problems. Thus, our claim is not that recent work is absent, but rather that structural uncertainty at the function level remains largely unexplored.
>
> [i]Ljung, L. and Glad, T. “On global identifiability for arbitrary model parametrizations.” Automatica, 30(2):265–276, 1994.
>
> [ii]Miao, H., Xia, X., Perelson, A. S., and Wu, H. “On identifiability of nonlinear ODE models and applications in viral dynamics.” SIAM Review, 53(1):3–39, 2011.
>
> [iii]Meshkat, N. and Sullivant, S. “Identifiable reparametrizations of linear compartment models.” Journal of Symbolic Computation, 63:46–67, 2014.
>
> [iv]Meshkat, N., Rosen, Z., and Sullivant, S. “Algebraic tools for the analysis of state space models.” In The 50th Anniversary of Gröbner Bases, Advanced Studies in Pure Mathematics, 77:171–205, 2018.
>
> Note that recent agent-based or LLM-driven approaches to scientific discovery face the same fundamental limitation: structural uncertainty cannot be resolved purely from data. From this perspective, our framework provides a complementary diagnostic tool for such automated scientific systems by explicitly identifying non-uniqueness through the structure of the governing equations. We will add this perspective to the Discussion.
>
>
> **[2. Redundancy in uncertainty description]**
>
> We agree that the explanation of uncertainty types was redundant. This was originally included for clarity based on prior submissions; however, since the distinction has been clearly understood by the reviewers, we will simplify the presentation. We will revise the Introduction as follows:
>
> Before: “We consider three types of uncertainty: structural uncertainty, model-form uncertainty, and data uncertainty…”
>
> After: “We focus on structural uncertainty, i.e., non-uniqueness that persists even with infinite, noiseless data when multiple model structures are consistent with the same observations.”
>
>
> **[3. Scope and generalization]**
>
> Our framework applies to PDEs that are linear in the derivatives of the coefficient function as a minimal setting to clearly isolate the core mechanism of structural uncertainty. Importantly, this setting already covers a nontrivial class of practical problems: the framework applies whenever the PDE is linear in coefficient derivatives, even if it is nonlinear in the observed variables. Both the diffusion equation and the wave-kinetic equation in the paper fall into this category. Both the diffusion equation and the wave-kinetic equation in the manuscript fall into this category. Also, Burgers-type equations, such as
> $u_t + u\\,u_x = \\nu(x)\\,u_{xx},$
> and nonlinear diffusion-type equations, such as
> $u_t = D(x)\\,u_{xx} + f(u),$
> are additional examples. We will clarify these extension pathways in the Discussion.

---

> > ### Author Rebuttal · Reviewer_H9rF · 2026-04-02
> >
> > Thank you for the answering the questions raised. I am convinced with the answers and have no further Questions.

---

> > > ### Author Response · Authors · 2026-04-05
> > >
> > > Thank you very much for your careful reading and for confirming that your concerns have been fully addressed.

---

### Official Review · Reviewer_WsCC · 2026-03-12

**Soundness:** 3
**Presentation:** 4
**Significance:** 3
**Originality:** 3
**Overall Recommendation:** 3
**Confidence:** 4

**Summary:**

The paper is organized around the idea of quantifying structural uncertainty in the estimation of PDE coefficients within Physics informed Machine Learning. The paper shows that naive predictive learning leads to non-unique identification of coefficients in a MHD setting which can be removed by using physics aware geometric constraints using a diagnostic rank based criterion on a derived matrix from the PDEs.

**Compliance With Llm Reviewing Policy:**

Affirmed.

**Key Questions For Authors:**

Kindly address the three weaknesses -- scope limitation, AI contribution and stress testing benchmarks.

**Limitations:**

Yes

**Strengths And Weaknesses:**

#### Strengths

1. The paper is extremely well written and well motivated. The identifiability issue in inverse problems using PIML is well developed and the treatment of UQ is quite well deliberated. In fact the framework of diagnosing identifiability is a strong contribution to scientific ML.

2. Within its scope - the theory is solid and well aligned with the paper. Although the logic itself is not novel or technical, it is developed without any handwavyness and provides an interpretable diagnostic tool. This is a strong part of the paper.

3. The wave kinetics example is a great test-example to not only understand the identifiability issue in scientific ML but also see the methodology in action.

#### Weaknesses
1. My main issue is that I don't think this is an ICML paper. First the theory is really scope limited and that is not even scratching the surface for this problem in scientific ML. The typical use case of (inverse problems in) sciML is field inversion, coupled physics (even one way coupling can introduce dependence of state on coefficients), unknown closure. In all of these, I don't know how to extend the current  rank based diagnostic that the paper has proposed.

2. AI contribution: For an application based paper, the AI contribution should be enough to stand on its own. The methodology is well motivated but I don't see a specific AI contribution that I think will make this suitable for ICML even though I agree that the problem itself is important but even then see point 1 above.

3. The benchmarks are mostly designed as an illustration for the theoretical result. This is great but doesn't quite lift itself to the benchmark territory. Can we stress test the framework in which we have noisy data, partial observability or distributional shift? Also there are no comparisons with some UQ aware PINNs; I understand these methods are not exactly uncovering structural uncertainty but at least a discussion or a basic comparison so that the readers can see that, say Bayesian PINNs or other such method give a posterior but it  is hard to isolate the structural UQ part.

---

> ### Author Rebuttal · Authors · 2026-03-30
>
> We sincerely thank the reviewer for the careful and constructive feedback, and for recognizing the clarity, motivation, and value of the identifiability diagnosis framework.
>
> We address the three main concerns below.
>
>
> **[1 Scope limitation]**
>
> The current framework assumes PDEs that are linear in the derivatives of the target function. Importantly, the PDE may still be nonlinear in the observed variables. This already covers a practically relevant class of physical PDEs. In addition to the diffusion equation and the wave-kinetic example in the paper, Burgers-type equations and diffusion equations with nonlinear terms also fall into this class.
>
> The basic idea also extends to systems with multiple PDEs. For example, the canonical equations of motion discussed in the paper form a coupled system. In this sense, coupled systems and field inversion problems may be approached by incorporating additional governing equations as constraints in the matrix. Closure-type models may also be treated by regarding closure terms as effective coefficients or additional equations. We will clarify these extension paths in the Discussion.
>
>
> **[2 AI contribution]**
>
> This work does not propose a new predictive model. Rather, it identifies a fundamental issue inherent in applying AI to physics through PIML and proposes a mathematical diagnostic framework to mitigate it. In this sense, the work contributes to AI research in a way similar to interpretable AI and theoretical analyses of AI, by analyzing the mechanism of PIML itself and deepening our understanding of it.
>
> In particular, the structural uncertainty studied here is not estimated from data, but is intrinsic to the governing PDE itself. A canonical example is gauge invariance of the vector potential in Maxwell’s equations. Since such uncertainty may involve infinite-dimensional degrees of freedom associated with arbitrary functions, it is in principle extremely difficult to identify from data alone.
>
> For this reason, the problem is relevant not only to PIML but more broadly to AI for Science. Rather than proposing a new predictor, this work provides a framework for diagnosing the validity of physical structures inferred by AI that cannot be fully captured from data alone. The framework therefore serves existing AI-based methods for physics, including PIML, as a tool for model validation and evaluation under uncertainty. We will clarify this positioning more explicitly in the Introduction.
>
>
> **[3 Stress testing, benchmarks, and UQ comparison]**
>
> As pointed out, the experiments in this work are designed not as benchmark evaluations but as illustrative demonstrations to isolate structural uncertainty. Their purpose is to show that evaluating structural uncertainty in advance can address problems that cannot be resolved by predictive loss alone.
>
> Structural uncertainty is intrinsic to the governing PDE and independent of data, whereas data noise, partial observability, and distribution shift give rise to data uncertainty and model-form uncertainty. As summarized in Table 1, it is important to evaluate these uncertainty types separately even when they coexist. Since structural uncertainty is independent of data, it can in principle be isolated even in such mixed settings. We therefore view the present framework as a first step toward disentangling different sources of uncertainty.
>
> In the wave-kinetic demonstration, the generated data already contain some numerical errors from simulation. In this sense, the demonstration can be viewed as a case where structural uncertainty is reduced in the presence of data uncertainty. While the structural uncertainty itself is not altered by the noise, the result suggests the practical importance of reducing structural uncertainty even when data uncertainty is present. We will add this clarification to the Discussion.
>
> We also agree that discussion of UQ-aware PINNs, such as Bayesian PINNs, is valuable. Structural uncertainty often corresponds to global, infinite-dimensional degrees of freedom, such as gauge invariance. In the wave-kinetic example, the Poisson bracket structure in Eq. (5) implies invariance under the addition of an arbitrary function. By contrast, Bayesian PINNs quantify uncertainty through finite-dimensional approximations such as Laplace approximations, variational inference, or sampling. It is therefore generally difficult for them to explicitly disentangle such infinite-dimensional non-uniqueness. In this sense, UQ methods and our framework are complementary: UQ methods quantify uncertainty, whereas our framework isolates and diagnoses its structural component. We will add this discussion to the Discussion section.

---

> > ### Author Rebuttal · Reviewer_WsCC · 2026-04-03
> >
> > Thank you for the detailed answers. I appreciate the formalism of the paper but I don't see it yet applicable broadly to AI for science. I still don't see the practical utility of the method beyond the setup in the paper. The benchmark comparisons to current UQ methods should empirically show where the proposed (rank formalism) shines! This is a great idea that needs a thorough comparison (AI based UQ methods) to stand out.

---

> > > ### Author Response · Authors · 2026-04-05
> > >
> > > Thank you for this helpful comment! We understand that, for an ML paper, comparison with existing methods is important, and following your suggestion, we conducted an additional comparison with B-PINNs on the wave-kinetic system.
> > >
> > > Our position is that, while Bayesian methods such as B-PINNs quantify posterior uncertainty, they do not explicitly disentangle whether that uncertainty is structural, data-driven, or model-form, nor do they readily represent functionally extended degrees of freedom in practice. Indeed, when we applied B-PINNs to the wave-kinetic system, the credible intervals of the estimated Hamiltonian remained finite and did not explicitly recover the theoretically known infinite-dimensional structural ambiguity $H(x,k_x,t)=H_{\mathrm{true}}(x,k_x,t)+f(I(x,k_x,t))$. An anonymous supplementary figure is provided here for reference: https://d.tmpfile.link/public/2026-04-05/8d6a0fdb-0268-4c04-bb43-e0eea9250995/fig_for_reviewerWsCC.pdf.
> > >
> > > To make this point concrete, we constructed an approximate procedure for estimating the “degrees of freedom of structural uncertainty” from the B-PINNs posterior. Specifically, we computed the covariance matrix of $P(H(x^{(1)},k_x^{(1)}), H(x^{(2)},k_x^{(2)}), \dots, H(x^{(N)},k_x^{(N)}) \mid \mathrm{Data})$, and used the number of large singular values as an approximation to the uncertainty degrees of freedom. This quantity corresponds to the degrees of freedom of the Hamiltonian function $H(x,k_x)$ itself, whereas the proposed method directly evaluates the degrees of freedom of $\partial_x H$ and $\partial_{k_x}H$.
> > >
> > > The comparison results are summarized below.
> > >
> > > | Methods | W/O constraints | W/ constraints | case of $I(x,k_x,t)=0$ |
> > > |---|---:|---:|---:|
> > > | Proposed | $\geq N_xN_{k_x}=10000$ | $0$ | $2N_xN_{k_x}=20000$ |
> > > | B-PINNs ($\mathrm{threshold}=\infty$) | $0 \pm 0.0$ | $0 \pm 0.0$ | $0 \pm 0.0$ |
> > > | B-PINNs ($\mathrm{threshold}=1.0$) | $0.67 \pm 0.47$ | $0.67 \pm 0.47$ | $9990.0 \pm 0.47$ |
> > > | B-PINNs ($\mathrm{threshold}=0.01$) | $9.5 \pm 0.96$ | $2.00 \pm 0.0$ | $9999.0 \pm 0.47$ |
> > >
> > > For the proposed method, the values are taken from the main text. For $I(x,k_x,t)=0$, they follow from $M$ being the zero matrix. For B-PINNs, we used 10000 posterior samples for this case. Since the Hamiltonian itself is theoretically unconstrained when $I(x,k_x,t)=0$, the expected degrees of freedom of $H(x,k_x)$ are $N_xN_{k_x}=10000$, and the B-PINNs results at threshold 1.0 and 0.01 are consistent with this expectation.
> > >
> > > These results show that only the proposed method consistently gives estimates that are theoretically consistent with the structural degrees of freedom across all tested cases. In contrast, the estimated degrees of freedom from B-PINNs depend strongly on the post hoc threshold used to extract dominant covariance modes, indicating that these “degrees of freedom” are not directly identified but depend on an additional post-processing criterion.
> > >
> > > This difference is consistent with the known structural ambiguity $H(x,k_x,t)=H_{\mathrm{true}}(x,k_x,t)+f(I(x,k_x,t))$. For the partial derivatives, $\partial_x H=\partial_x H_{\mathrm{true}}+\partial_I f\,\partial_x I$ and $\partial_{k_x} H=\partial_{k_x} H_{\mathrm{true}}+\partial_I f\,\partial_{k_x} I$, so the ambiguity in $\partial_x H$ and $\partial_{k_x}H$ is governed by a single functional degree of freedom $g(I):=\partial_I f$. In the worst case, when $I(x,k_x,t)$ takes distinct values on all grid points, In the worst case, when $I(x,k_x,t)$ takes distinct values on all grid points, this yields at least $N_xN_{k_x}$ structurally distinguishable degrees of freedom in the discretized setting, which is consistent with the lower bound given by the proposed method.
> > >
> > > By contrast, B-PINNs represent posterior uncertainty over $H(x,k_x)$ itself. If the functional ambiguity $f(I(x,k_x,t))$ were explicitly captured, one would expect covariance modes consistent with this degree of freedom. However, this was not observed in practice. Moreover, under constraints, the proposed method correctly returns zero structural degrees of freedom, whereas B-PINNs still leave a small but nonzero estimate, making it unclear whether the remaining spread reflects residual structural ambiguity or merely finite-sample / approximation effects.
> > >
> > > Overall, this comparison suggests that B-PINNs and the proposed rank-based framework are complementary rather than interchangeable: B-PINNs quantify posterior uncertainty, whereas the proposed method directly diagnoses the structural origin and degrees of freedom of non-uniqueness from the PDE itself. We will add the corresponding figures and tables to the section of Demonstration, Discussion, and Appendix in the revised manuscript.

---

### Official Review · Reviewer_NxRE · 2026-03-13

**Soundness:** 3
**Presentation:** 3
**Significance:** 3
**Originality:** 4
**Overall Recommendation:** 5
**Confidence:** 4

**Summary:**

This paper studies structural uncertainty in physics-informed inverse problems, arguing that predictive accuracy alone can mask non-uniqueness in estimated PDE coefficient functions. It proposes a three-step framework: diagnose structural uncertainty through the rank of a matrix derived from the PDE structure, introduce physically motivated constraints, and analyze how estimated solutions vary with constraint strength rather than selecting a single model by predictive performance. The paper demonstrates the framework on several idealized systems and on a wave-kinetic-equation example motivated by fusion-related turbulence modeling.

**Compliance With Llm Reviewing Policy:**

Affirmed.

**Key Questions For Authors:**

Can the authors more clearly state the computational cost and numerical requirements of diagnosing structural uncertainty in realistic PDE systems?

How broadly does the rank-based diagnosis extend to nonlinear, partially observed, or noisy real-world settings?

Can the authors provide a more concrete recipe for choosing physically induced constraints in domains where domain knowledge is weak or contested?

What would be the simplest practical workflow for a PIML practitioner who wants to apply this framework to a new inverse problem?

**Limitations:**

The paper’s limitations are not fatal, but they should be stated more plainly: much of the evidence is demonstrative, the framework may be expensive or difficult to instantiate in complex systems, and the paper does not yet show large-scale end-to-end integration into modern PIML pipelines.

**Strengths And Weaknesses:**

Strengths
This is a thoughtful paper with a strong conceptual contribution. It makes an important distinction between structural, model-form, and data uncertainty, and convincingly argues that structural uncertainty should be diagnosed before applying standard uncertainty-quantification tools. That framing is valuable well beyond the specific examples in the paper.

The proposed three-step framework is clear and principled. I especially liked the emphasis that one should not pick a constraint strength purely by predictive accuracy, but instead inspect the family of physically plausible solutions. That is a nonstandard but scientifically sensible stance.

The demonstrations also support the central message: the paper shows cases where predictive loss remains similar even though structural uncertainty differs substantially, reinforcing the claim that standard predictive evaluation is insufficient.

Weaknesses
The main weakness is that the paper is more of a framework-and-position paper than a fully developed ML method paper. The theoretical setup appears meaningful, but the practical payoff for typical ICML readers may feel limited unless the authors sharpen exactly when the framework is tractable and how broadly it applies beyond the chosen examples.

A second issue is that the empirical evaluation seems oriented toward illustrative demonstrations rather than strong benchmark-style validation. That is acceptable for this kind of paper, but then the paper should more explicitly position itself as providing conceptual and diagnostic tools rather than state-of-the-art predictive performance.

I also wanted more detail about the actual computational cost and numerical stability of the proposed diagnosis step in realistic high-dimensional systems. The wave-kinetic example is interesting, but I am not fully convinced yet that the framework is easy to use in broad practice.

---

> ### Author Rebuttal · Authors · 2026-03-30
>
> We sincerely thank the reviewer for the insightful and constructive questions. We are also grateful that the reviewer recognized the value of our framework as a conceptual and diagnostic tool for understanding structural uncertainty, rather than merely as a predictive benchmark. This perspective is closely aligned with the main purpose of our work.
>
> To clarify the positioning of this paper more explicitly, we will revise the Introduction to state this point more clearly.
>
> In the Introduction, we will add the following sentence:
> “This work is intended as a conceptual and diagnostic framework for assessing structural uncertainty in physics-informed inverse problems, rather than as a benchmark-oriented predictive method.”
>
>
> **[Q1: Computational cost and numerical requirements]**
>
> We thank the reviewer for raising this important point. Our method evaluates structural uncertainty analytically through the matrix $M$ derived from the PDE structure. In grid-based settings, $M$ is typically sparse because each grid-point equation involves only local differential terms. As a result, both the construction of $M$ and the subsequent linear-algebraic operations required for rank computation are substantially cheaper than in the dense case.
>
> At the same time, we agree that practical numerical settings require careful interpretation. When $M$ is constructed from finite or noisy observations, the computed rank may no longer represent purely structural uncertainty; instead, it may reflect a mixture of structural, data, and model-form uncertainties. We will clarify this point in the revised manuscript.
>
>
> **[Q2: Applicability to nonlinear, partially observed, and noisy settings]**
>
> We thank the reviewer for pointing out the importance of clarifying the scope of applicability. The proposed method applies to PDEs that are linear in the derivatives of the target coefficient function, including cases in which the observed variables themselves enter the governing equations nonlinearly. In addition, the framework can be extended to partially observed systems when governing equations for the latent variables are available. Regarding noisy settings, noise primarily contributes to data uncertainty rather than structural uncertainty, and a full treatment of finite-sample or noisy inference lies outside the central scope of the present work.
>
> To avoid misunderstanding, we will state this scope and limitation more explicitly in the revised Discussion.
>
>
> **[Q3: Choosing physically induced constraints]**
>
> We appreciate this insightful question, as it concerns one of the main practical implications of our framework. In our view, structural uncertainty represents intrinsic degrees of freedom of the physical system, such as gauge freedom, and domain knowledge typically constrains only part of that freedom. For this reason, we do not advocate selecting a single “correct” constraint a priori. Rather, we believe the more appropriate strategy is to explore multiple physically plausible constraints and examine the corresponding family of admissible solutions.
>
> We will revise the manuscript to make this philosophy clearer.
>
>
> **[Q4: Practical workflow]**
>
> We also thank the reviewer for encouraging us to make the practical use of the framework more explicit. A simple workflow is:
> (1) construct the matrix $M$,
> (2) diagnose structural uncertainty via rank analysis,
> (3) introduce candidate physically motivated constraints,
> (4) sweep the constraint strength, and
> (5) analyze the resulting family of solutions.
>
> We agree that stating this workflow explicitly would improve the paper, and we will incorporate it into the revised Discussion.

---

> > ### Author Rebuttal · Reviewer_NxRE · 2026-04-07
> >
> > The authors agreed to address my concerns in the revised manuscript.

---

> > > ### Author Response · Authors · 2026-04-08
> > >
> > > Thank you very much for your great support and for confirming that your concerns have been fully addressed.

---

### Official Review · Reviewer_MMWD · 2026-03-16

**Soundness:** 2
**Presentation:** 3
**Significance:** 3
**Originality:** 3
**Overall Recommendation:** 3
**Confidence:** 3

**Summary:**

The authors propose a rank-based diagnostic for structural uncertainty in physics-informed coefficient function estimation. While the problem is genuinely important, the theoretical framework requires the continuum limit ($\varepsilon \to 0$) and PDEs linear in the coefficient derivatives, and the paper does not address the gap between these assumptions and practice. The main experiment achieves a cosine similarity of only $0.45 \pm 0.29$ (Table 2), which is difficult to reconcile with the theoretical uniqueness guarantee. No comparison with existing UQ methods is provided. See the detailed comments below.

**Compliance With Llm Reviewing Policy:**

Affirmed.

**Key Questions For Authors:**

See above

**Limitations:**

See above

**Strengths And Weaknesses:**

* Table 2: cosine similarity is $0.45 \pm 0.29$ with constraints over 30 trials. Since the theory predicts uniqueness up to a constant after imposing the symmetry constraint, and the authors correct for the constant (p. 8, line 434), similarity should be close to 1.0. The fact that it is 0.45 with standard deviation 0.29 indicates severe practical failure. Is this caused by optimization difficulties, finite data, or the soft penalty formulation? This is the most important result in the paper, and it calls into question the framework's practical value.

* Theorem 1 is stated for $\varepsilon \to 0$. Unfortunately, the paper does not study how rank($\mathbf{M}$) behaves as $\varepsilon$ increases, nor do the appendix demonstrations (B, C, D) systematically vary grid spacing. If the rank condition holds only in the continuum limit, its practical value is limited to settings where the grid is fine enough to approximate the continuum, which is the regime where structural uncertainty may already be detectable through simpler means, e.g., training multiple models and comparing solutions.

* The unconstrained case achieves lower prediction loss ($2.58 \times 10^{-8}$ vs. $5.46 \times 10^{-8}$). While the authors correctly present this as evidence that predictive loss is insufficient, it also means the constraint worsens data fit by approximately $2\times$. No $\lambda$ sweep is shown despite Step 3 of the framework being explicitly devoted to analyzing constraint strength dependence. This is a considerable omission: Step 3 is the most practically relevant component, and the main experiment does not demonstrate it.

* The paper does not compare against any existing UQ method. B-PINNs (Yang et al., 2021), cited in Section 2.3, provide posterior distributions over the coefficient function. Running B-PINNs on the wave-kinetic problem and checking whether the posterior spread reveals the same non-uniqueness would clarify the added value of the rank formalism. This comparison is essential, and its absence is a considerable weakness.

* The Hamiltonian example shows rank($\mathbf{M}$) = $2Nd$, confirming that $H(q,p)$ is uniquely determined up to a constant. However, this is tantamount to the classical result that Hamilton's equations fully determine the Hamiltonian given complete phase-space trajectories. Similarly, the Lagrange rank deficiency restates the well-known gauge freedom resolvable via the Legendre transform. The one-dimensional diffusion example derives the additive-constant ambiguity through a considerable amount of algebra, when the result follows in two lines from elementary ODE theory. While the unified formalism has value, the paper should clearly state what Theorem 1 reveals beyond what the classical theory of each specific system already establishes.

* Definition 1 is tantamount to the classical notion of non-uniqueness for inverse problems. In a Bayesian framework, structural uncertainty would manifest as multimodality or broad support in $p(a \mid \text{data})$, characterizing not only *whether* non-uniqueness exists but also its degree and structure. While the null space of $\mathbf{M}$ contains this information in principle, the paper never exploits it, rendering the diagnostic binary rather than quantitative.

* Equation (8) imposes symmetry as a soft $L_2$ penalty. If the symmetry holds exactly, a natural question is why it is not enforced as a hard architectural constraint, e.g., restricting the neural network to produce even functions of $k_x$. A soft penalty allows violations, introducing a solution family parameterized by $\lambda$ that the rank analysis does not account for.

* Remark 1 notes that lower-order derivative terms provide additional constraints that may reduce structural uncertainty, but this "lifting" is left to future work. This omission directly affects the wave-kinetic analysis, since equation (5) involves both $\partial_x H$ and $\partial_{k_x} H$, which according to Remark 1 should provide information about $H$ itself through cumulative sums. Accounting for this could change the conclusions of Section 5 regarding the necessity of the symmetry constraint.

* The paper assumes noiseless data and known PDE structure. In practice, data noise can mask or mimic structural non-uniqueness by perturbing the entries of $\mathbf{M}$. The clean separation of uncertainty types in Table 1, while conceptually appealing, may not hold when these types interact.

---

> ### Author Rebuttal · Authors · 2026-03-31
>
> We sincerely thank the reviewer for the detailed and thoughtful feedback. The comments not only helped us clarify important aspects of our work, but also opened up new and exciting directions for future research.
>
> We address the key points below.
>
>
> [1. Cosine similarity and practical performance]
>
> In our case, data accuracy is limited because simulation-based generation introduces spatially local numerical errors, making exact Hamiltonian recovery difficult. More generally, discretization, numerical differentiation, and model approximation introduce noise even in simulation data. By contrast, Theorem 1 diagnoses structural uncertainty inherent in the PDE and is independent of data. Thus, this experiment does not aim at perfect reconstruction, but to show that explicitly accounting for structural uncertainty enables physically consistent solutions. Indeed, unconstrained models achieve lower prediction loss yet select physically incorrect solutions, highlighting that predictive accuracy alone is insufficient. This gap between predictive performance and physical correctness motivates our framework.
>
>
> [2. Continuum limit and discretization]
>
> Structural uncertainty is an intrinsic property of the PDE and is independent of discretization or data. The rank in our framework is a theoretical quantity characterizing the structure of the governing equation, not a numerically estimated value.
> Learning-based approaches cannot fully recover structural uncertainty, as they select a single solution from a non-unique solution space. In contrast, our method directly analyzes the null space of a matrix derived from the PDE, allowing the number and structure of non-uniqueness to be identified analytically.
> This distinction is essential: even in practical finite-grid settings, diagnosing structural uncertainty is necessary for reliable model identification.
>
>
> [3. λ sweep (Step 3)]
>
> We have conducted a systematic λ-sweep experiment as follows.
>
> | $\lambda$ | L2-loss (mean$\pm$ std) | cosine similarity (mean$\pm$ std) |
> |------|----------|----------|
> | 100   |5.32e-07$\pm$7.81e-07|0.19$\pm$0.28|
> | 10   |1.27-06$\pm$2.27e-06|0.18$\pm$0.24|
> | 1   | 1.84e-07$\pm$6.88e-07|0.49$\pm$0.26|
> | 0.1   |3.79e-08$\pm$5.23e-09|0.59$\pm$0.15|
> | 0.01   |2.67e-08$\pm$1.85e-09|0.46$\pm$0.26|
> | 0.001   |2.10e-08$\pm$1.66e-09|0.31$\pm$0.16|
>
> The results show:
> (1) L2 loss decreases as constraint strength increases.
> (2) Cosine similarity peaks at an intermediate λ.
> These results confirm that predictive loss alone is insufficient and that model selection must consider the structure of the solution space.
> Moreover, in realistic settings with noise and model limitations, symmetry may not hold exactly; thus, exploring the solution family via λ-sweep is crucial for gaining physical insight.
>
>
> [4. Comparison with UQ methods]
>
> UQ methods (e.g., Bayesian PINNs) estimate uncertainty via posterior distributions but cannot clearly disentangle structural, data, and model uncertainties.
> Structural uncertainty often corresponds to infinite-dimensional degrees of freedom (e.g., gauge invariance), which cannot be fully represented by finite-dimensional approximations.
> Therefore, our method is complementary: UQ quantifies uncertainty magnitude, while our approach identifies its structural origin through the PDE.
>
>
> [5. Relation to classical results]
>
> The examples are illustrative and not intended as novel results. Our contribution is a unified framework that characterizes non-uniqueness through linear dependence and null space analysis, providing a systematic connection between classical theory and data-driven modeling.
>
>
> [6. Binary vs. quantitative diagnosis]
>
> Our method is not limited to binary diagnosis. The null space provides the number and structure of degrees of freedom, enabling a quantitative and interpretable characterization of non-uniqueness.
>
>
> [7. Soft vs. hard constraints]
>
> Hard constraints can be overly restrictive in finite, noisy settings. We observe performance degradation at large λ.
> Soft constraints allow continuous analysis of the solution space, and λ-sweep serves as a bridge between ideal structural uncertainty and practical learning behavior.
>
>
> [8. Remark 1]
>
> Remark 1 indicates that lower-order terms can provide additional constraints. However, Eq. (5) does not include $H$ itself, only $\partial_x H$ and $\partial_{k_x} H$.
> Thus, such constraints are not present in this setting, and our conclusion that structural uncertainty can persist remains unchanged.
>
>
> [9. Noise and interaction of uncertainties]
>
> As pointed out, evaluating interacting uncertainties is highly challenging. Our goal is to isolate structural uncertainty, which is intrinsic to the PDE and independent of data.
> While model and data uncertainties are more difficult to separate, their interaction may lead to additional behaviors.
> Analyzing such interactions is an important and promising future direction, and we thank the reviewer for this insightful and stimulating suggestion.

---

> > ### Author Rebuttal · Reviewer_MMWD · 2026-03-31
> >
> > The $\lambda$-sweep in response to [3] is appreciated and partially addresses that concern. The core issues, however, remain. The rebuttal to [1] reframes the low cosine similarity as intentional, but does not explain why a result the theory predicts should be near 1.0 is observed at 0.45 ± 0.29; this tension is unresolved. The response to [4] argues philosophically against the B-PINNs comparison rather than providing it, leaving the added value of the rank formalism undemonstrated empirically. These two concerns are central to evaluating practical utility and are not easily addressed in a short rebuttal.

---

> > > ### Author Response · Authors · 2026-04-05
> > >
> > > Thank you for this important follow-up comment. We respond to two points below.
> > >
> > > **[On the Hamiltonian estimation error in the wave-kinetic example]**
> > >
> > > We understand your question as asking for concrete evidence that the Hamiltonian estimation error mainly comes from localized numerical noise in the generated data. To address this, we performed an additional experiment.
> > >
> > > Under the symmetry constraint, we computed $M^{-1}$ and estimated $\partial_x H$ and $\partial_{k_x}H$ from Eq. (17) as a linear problem. In Fig. 2, wave-like fine structures appear in phase space in the regions $x=-3\sim0$ and $x=6\sim8$, which we interpret as numerical artifacts caused by simulation error. We therefore divided the domain into a noisy area and a noise-free area, and compared the estimation errors of the Hamiltonian derivatives.
> > >
> > > | Estimation parameters | RMSE of noise-free area (mean $\pm$ std) | RMSE of noisy area (mean $\pm$ std) |
> > > |---|---:|---:|
> > > | $\partial_x H$ | $0.197 \pm 0.033$ | $0.389 \pm 0.010$ |
> > > | $\partial_{k_x} H$ | $0.285 \pm 0.051$ | $0.382 \pm 0.0234$ |
> > >
> > > The error is clearly larger in the noisy area, supporting our interpretation that numerical error is a major cause of the degradation in Hamiltonian recovery. Thus, the low cosine similarity reflects reduced recoverability due to localized numerical noise rather than a failure of the structural diagnosis itself (anonymous figure: https://d.tmpfile.link/public/2026-04-05/15056d6d-a055-48c4-968a-980cb589aef1/fig1_for_reviewerMMWD.pdf).
> > >
> > > **[On the comparison with B-PINNs]**
> > >
> > > We agree that our previous rebuttal was too conceptual and did not provide an empirical comparison with B-PINNs. Following your suggestion, we conducted an additional comparison on the same wave-kinetic system.
> > >
> > > B-PINNs quantify posterior uncertainty, but do not explicitly disentangle whether that uncertainty is structural, data-driven, or model-form, nor do they readily represent functionally extended degrees of freedom in practice. When we applied B-PINNs to the wave-kinetic system, the credible intervals of the estimated Hamiltonian remained finite and did not explicitly recover the known structural ambiguity $H(x,k_x,t)=H_{\mathrm{true}}(x,k_x,t)+f(I(x,k_x,t))$ (anonymous figure: https://d.tmpfile.link/public/2026-04-05/698d1b72-d669-4b87-9331-c9d4ac2ba530/fig2_for_reviewerMMWD.pdf). The step size was tuned so that the average acceptance rate remained in the range $0.7$--$0.95$.
> > >
> > > To make this concrete, we estimated the “degrees of freedom of structural uncertainty” from the B-PINNs posterior by computing the covariance matrix of $P(H(x^{(1)},k_x^{(1)}),\dots,H(x^{(N)},k_x^{(N)})\mid\mathrm{Data})$ and counting large singular values. This quantity corresponds to the degrees of freedom of $H(x,k_x)$ itself, whereas the proposed method diagnoses those of $\partial_x H$ and $\partial_{k_x}H$.
> > >
> > > | Methods | W/O constraints | W/ constraints | case of $I(x,k_x,t)=0$ |
> > > |---|---:|---:|---:|
> > > | Proposed | $\geq N_xN_{k_x}=10000$ | $0$ | $2N_xN_{k_x}=20000$ |
> > > | B-PINNs ($\mathrm{threshold}=\infty$) | $0 \pm 0.0$ | $0 \pm 0.0$ | $0 \pm 0.0$ |
> > > | B-PINNs ($\mathrm{threshold}=1.0$) | $0.67 \pm 0.47$ | $0.67 \pm 0.47$ | $9990.0 \pm 0.47$ |
> > > | B-PINNs ($\mathrm{threshold}=0.01$) | $9.5 \pm 0.96$ | $2.00 \pm 0.0$ | $9999.0 \pm 0.47$ |
> > >
> > > For the proposed method, the values are taken from the main text. For $I(x,k_x,t)=0$, they follow from $M$ being the zero matrix. For B-PINNs, we used 10000 posterior samples for this case. Since the Hamiltonian itself is theoretically unconstrained when $I(x,k_x,t)=0$, the expected degrees of freedom of $H(x,k_x)$ are $N_xN_{k_x}=10000$, and the B-PINNs results at threshold 1.0 and 0.01 are consistent with this expectation.
> > >
> > > The key observations are: (i) the proposed method remains theoretically consistent across all tested cases, whereas the B-PINNs estimate depends strongly on the post hoc threshold; and (ii) in the unconstrained wave-kinetic case, the known functional ambiguity $f(I)$ does not appear explicitly in the B-PINNs posterior structure. Under constraints, the proposed method correctly returns zero structural degrees of freedom, whereas B-PINNs still leave a small but nonzero estimate, making it unclear whether the residual spread reflects structural ambiguity or finite-sample / approximation effects.
> > >
> > > Overall, these results suggest that B-PINNs and the proposed rank-based framework are complementary rather than interchangeable: B-PINNs quantify posterior uncertainty, whereas the proposed method directly diagnoses the structural origin and degrees of freedom of non-uniqueness from the PDE itself. From a Bayesian viewpoint, structural uncertainty may induce highly extended flat directions, which may require more advanced yet practical inference schemes. We will add the corresponding figures, tables, and discussion to the Demonstration section, the Discussion section, and the Appendix in the revised manuscript.

---

### Decision · Program_Chairs · 2026-04-30

**Decision:**

Reject

**Comment:**

This paper addresses an important problem in physics-informed inverse problems: low predictive or PDE residual loss does not guarantee unique or physically meaningful recovery of coefficient functions. Reviewers generally found the paper well written, clearly motivated, and conceptually valuable as a diagnostic framing of “structural uncertainty” in PIML. In particular, the distinction among structural, model form, and data uncertainty, and the emphasis on diagnosing non-uniqueness before relying on predictive performance, were viewed as useful perspectives for AI for science practitioners.

At the same time, the central concern across the reviews is one of positioning and contribution type. Multiple reviewers noted that the paper reads more as a conceptual or diagnostic framework than as a standard ICML methods paper. The empirical section is largely demonstrative, and while that may be appropriate for a conceptual paper, it does not yet establish broad practical utility or benchmark-level evidence for typical ICML standards. Reviewers also raised concerns about the limited scope of the current theory, the lack of convincing stress tests in realistic settings, and the absence of sufficiently strong empirical comparison to uncertainty aware baselines such as Bayesian PINNs.

More fundamentally, I share the reviewers’ concern that the core issue identified here is closer to a classical inverse-problem well-posedness / identifiability / non-uniqueness question than to a new ML contribution. Indeed, the paper itself characterizes structural uncertainty as non-uniqueness that persists even with infinite noiseless data, and explicitly situates the problem within the literature on the ill-posedness of inverse problems. The proposed rank-based diagnosis is therefore best understood as a reformulation or repackaging of a classical inverse-problems concern within the PIML setting, rather than as a new methodological or theoretical advance in machine learning.

For these reasons, while I appreciate the paper’s clarity and the importance of the underlying problem, I do not find that it currently makes a sufficiently strong method or theory contribution for the main ICML track. The contribution appears primarily in framing and positioning, with most of the substantive novelty lying outside machine learning proper. In that sense, I believe this work would be better suited to a position/framework-oriented venue or track, where its main value as a conceptual warning and reframing for PIML can be more appropriately recognized. If the authors choose to resubmit this work to ML conference, I would encourage them to consider the position papertrack, where the paper’s main strength as a conceptual framing of an important issue in PIML would be a better fit.